# Shape-Informed Clustering of Multi-Dimensional Functional Data via Deep Functional Autoencoders

**Samuel  Singh**
School of Computer Science and Statistics
Trinity College Dublin

**Shirley  Coyle**
School of Electronic Engineering
Dublin City University

**Mimi  Zhang**
School of Computer Science and Statistics
Trinity College Dublin

## Abstract

We introduce `FAEclust`, a novel functional autoencoder framework for cluster analysis of multi-dimensional functional data, data that are random realizations of vector-valued random functions. Our framework features a universal-approximator encoder that captures complex nonlinear interdependencies among component functions, and a universal-approximator decoder capable of accurately reconstructing both Euclidean and manifold-valued functional data. Stability and robustness are enhanced through innovative regularization strategies applied to functional weights and biases. Additionally, we incorporate a clustering loss into the network's training objective, promoting the learning of latent representations that are conducive to effective clustering. A key innovation is our shape-informed clustering objective, ensuring that the clustering results are resistant to phase variations in the functions. We establish the universal approximation property of our non-linear decoder and validate the effectiveness of our model through extensive experiments.

## 1  Introduction

Advancements in information technology have enabled the collection of multi-dimensional functional data (FD) across various fields. Analyzing multi-dimensional FD is challenging due to its complexity and the intricate relationships between variables that change over time and space. The large volume of FD, especially in fields like medical imaging and climate science, further complicates efficient data processing. Zhang and Parnell [37] recently conducted a comprehensive review of clustering methods for FD. The review highlights that current approaches to clustering multi-dimensional FD typically follow one of two strategies: (1) performing multivariate functional principal component analysis [5] and subsequently applying traditional multivariate clustering techniques to the functional principal component score vectors, or (2) defining a (dis)similarity measure for multi-dimensional FD and then using a (dis)similarity-based clustering method. Although a few other works apply alternative basis systems rather than the eigen-functions of the covariance operator, they are still limited to learning linear representations of FD. Given that the relationships between dimensions can be nonlinear and complex, linear methods, including those derived from multivariate functional principal component analysis, often fall short in effectively learning FD.

Neural networks, known for their ability to model nonlinear relationships, offer a compelling alternative for multi-dimensional FD analysis. Rossi et al. [23] was among the first to apply multilayer perceptrons (MLPs) to FD by introducing functional weights in the first hidden layer. They demonstrated that functional MLPs can approximate continuous mappings from a compact subset of a functional space to $\mathbb{R}$ with arbitrary precision. In the context of function-on-function regression,

Wang et al. [33] converted FD into vectorial form using multivariate functional principal component analysis, subsequently building a traditional neural network for regression where both inputs and outputs were multivariate (i.e., functional principal component score vectors). Unlike [23], which modeled functional weights by cubic B-splines, Yao et al. [35] approximated each functional weight (namely their discrete evaluations over a grid) by a neural network. Thind et al. [32] extended the functional MLP framework to handle inputs that include both functional and multivariate data. Heinrichs et al. [8] defined each functional neuron to be a convolution $(w * y)$, where the functional weight $w$ is a translation-invariant kernel: $w(s, t) = w(s - t)$. A few recent works have developed neural network architectures for operator learning, namely learning mappings between two function spaces [16, 6, 24]. Notably, all these studies focus on supervised learning, designing neural networks for tasks like regression or classification and training neural networks with labeled data. As a result, they are not suited for cluster analysis, which requires identifying patterns in unlabeled data.

For cluster analysis of FD, a logical progression is to adapt the autoencoder architecture to FD. Hsieh et al. [10] made strides in this direction by developing a functional autoencoder (FAE), where the functional weights are modeled as integral kernels. However, this approach poses substantial computational challenges, as it requires training a separate integral kernel for each connection between neurons, making the training process exceedingly complex. Seidman et al. [25] applied variational autoencoders to FD. However, their approach has a critical limitation: the inputs (and outputs) are not continuous functions but rather function evaluations over a fixed grid. Therefore, the model is not discretization-invariant, and new architectures with new parameters may be needed to achieve the same error for data with varying discretization.

In this work, we introduce `FAEclust`, [1] a comprehensive Python framework for clustering multi-dimensional FD. `FAEclust` is capable of handling FD in both linear spaces and Riemannian manifolds, combining computational efficiency with robustness to phase variation (time warping) and high stability. Moreover, we prove that the functional decoder is a universal approximator. The paper is organized as follows: Section 2 details the `FAEclust` architecture, Section 3 introduces the training objective, and Section 4 presents the clustering methodology. Section 5 benchmarks `FAEclust` against state-of-the-art methods, and Section 6 discusses implications and future directions.

## 2    Functional autoencoder

In FD analysis, subjects are represented by random functions rather than traditional (multivariate) random variables. Below, we introduce the definitions and assumptions used throughout this work. Let $\{\boldsymbol{y}_i : i = 1, \ldots, n\}$ be a set of $n$ independent $p$-dimensional *sample functions*: $\boldsymbol{y}_i(t) = (y_i^1(t), \ldots, y_i^p(t))^T \in \mathbb{R}^p$ for any $t \in \mathcal{T}$, where $\mathcal{T}$ is a compact interval of $\mathbb{R}$; the superscript $T$ is the transpose operator. Each sample function $\boldsymbol{y}_i$ is a random realization of a $p$-dimensional *random function* $\vec{Y} = (Y^1, \ldots, Y^p)^T$. For $d = 1, \ldots, p$, the $n$ one-dimensional sample functions $\{y_1^d, \ldots, y_n^d\}$ are independent realizations of the component random function $Y^d = Y^d(t) = Y^d(t, \omega)$, defined on a probability space $(\Omega, \mathcal{F}, \mathrm{Pr})$ and taking values in $\mathcal{H}(\mathcal{T}, \mathbb{R})$. Here, $\mathcal{H}(\mathcal{T}, \mathbb{R})$ is the separable Hilbert space of all square-integrable measurable functions that are defined on $\mathcal{T}$ and taking values in $\mathbb{R}$. That is, $Y^d$ is a measurable map from $\Omega$ to $\mathcal{H}(\mathcal{T}, \mathbb{R})$. Alternatively, we can view the function value $y_i^d(t)$ as a realization of the random variable $Y^d(t, \cdot)$, a mapping from $(\Omega, \mathcal{F})$ to $(\mathbb{R}, \mathcal{B}_\mathbb{R})$, where $\mathcal{B}_\mathbb{R}$ is the Borel $\sigma$-algebra of $\mathbb{R}$. In the following, we will suppress the dependence of a random function $Y(\cdot, \omega)$ on $\Omega$ and simply write $Y$. Appendix A provides additional preliminaries on multi-dimensional FD.

To formally define the FD clustering problem, assume there are $K(\geq 2)$ clusters in the population. If a sample function $\boldsymbol{y}_i$ belongs to the $k$th cluster ($1 \leq k \leq K$), then it is a realization of the $k$th random function $\vec{Y}_k$, defined on the probability space $(\Omega, \mathcal{F}, \mathrm{Pr}_k)$. In other words, there are $K$ different probability measures defined on the $\sigma$-algebra $(\Omega, \mathcal{F})$. Given access only to the discrete evaluations $\{\boldsymbol{y}_i(t_{i1}), \ldots, \boldsymbol{y}_i(t_{ir_i})\}_{i=1}^n$, a clustering method is to identify the underlying random function for each sample function $\boldsymbol{y}_i$. Table 4 in Appendix A summarizes the notations we will use throughout the work. All vectors are column vectors.

---

[1] https://github.com/samuelveersingh/FAEclust

## 2.1 Network architecture

An FAE consists of an encoder $\mathcal{E}$ and a decoder $\mathcal{D}$; the encoder $\mathcal{E}$ maps a sample function $\boldsymbol{y} \in \mathcal{H}(\mathcal{T}, \mathbb{R}^p)$ into a latent representation $\boldsymbol{x} \in \mathbb{R}^s$, and the decoder $\mathcal{D}$ maps the representation $\boldsymbol{x}$ to a reconstruction of $\boldsymbol{y}$, denoted by $\hat{\boldsymbol{y}} \in \mathcal{H}(\mathcal{T}, \mathbb{R}^p)$. That is, the encoder is a mapping from the function space to a latent finite-dimensional space $\mathcal{E} : \mathcal{H}(\mathcal{T}, \mathbb{R}^p) \mapsto \mathbb{R}^s$, and the decoder is a mapping from the latent space back to the function space $\mathcal{D} : \mathbb{R}^s \mapsto \mathcal{H}(\mathcal{T}, \mathbb{R}^p)$. Cluster analysis is performed on $\{\mathcal{E}(\boldsymbol{y}_i)\}_{i=1}^n$, the embedded data in the latent space. Figure 1 presents a vanilla FAE architecture, where the *functional weights* $\{\mathrm{w}_{q,d}^{(1)} : 1 \leq q \leq q_1, 1 \leq d \leq p\}$ and $\{\omega_{d,q}^{(1)} : 1 \leq d \leq p, 1 \leq q \leq \tilde{q}_1\}$ are continuous functions, distinguishing them from typical scalar weights. Each node in the input

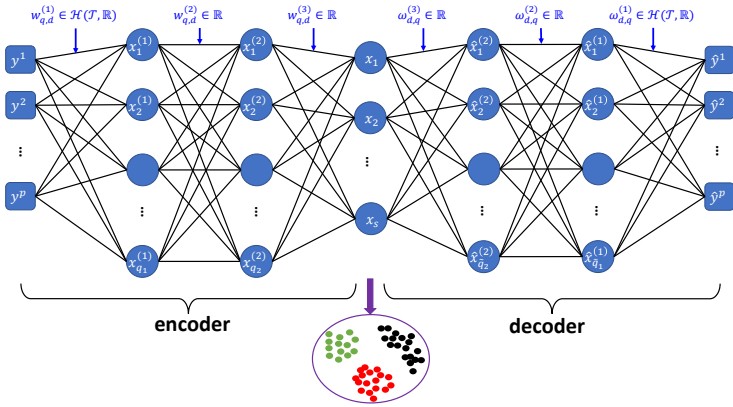

Figure 1: An illustrative FAE architecture, where we have five hidden layers.

layer accepts a function in $\mathcal{H}(\mathcal{T}, \mathbb{R})$, and in the output layer delivers a function in $\mathcal{H}(\mathcal{T}, \mathbb{R})$. The FAE architecture depicted in Figure 1 can be succinctly described by the following flow of information:

$$\boldsymbol{y} \xrightarrow{\mathrm{w}_{q,d}^{(1)} \in \mathcal{H}(\mathcal{T}, \mathbb{R})} \boldsymbol{x}^{(1)} \xrightarrow{\mathrm{MLP}} \boldsymbol{x} \xrightarrow{\mathrm{MLP}} \hat{\boldsymbol{x}}^{(1)} \xrightarrow{\omega_{d,q}^{(1)} \in \mathcal{H}(\mathcal{T}, \mathbb{R})} \hat{\boldsymbol{y}}.$$

Here, MLP stands for Multi-Layer Perceptron. This representation clearly demonstrates that the flow of information through the hidden layers, namely $\boldsymbol{x}^{(1)} \xrightarrow{\mathrm{MLP}} \boldsymbol{x} \xrightarrow{\mathrm{MLP}} \hat{\boldsymbol{x}}^{(1)}$, essentially forms a classical MLP autoencoder. Define weight matrices of appropriate dimensions:

$$\boldsymbol{W}^{(1)}(t) = [\mathrm{w}_{q,d}^{(1)}(t)] \in \mathbb{R}^{q_1 \times p}, \quad \text{and} \quad \boldsymbol{\mathcal{W}}^{(1)}(t) = [\omega_{d,q}^{(1)}(t)] \in \mathbb{R}^{p \times \tilde{q}_1}.$$

The feedforward equations for the first hidden layer and the output layer are given by:

$$\boldsymbol{x}^{(1)} = a\left(\int_{\mathcal{T}} \boldsymbol{W}^{(1)}(t)\boldsymbol{y}(t)dt + \boldsymbol{b}\right),$$
$$\hat{\boldsymbol{y}}(t) = \boldsymbol{\mathcal{W}}^{(1)}(t)\hat{\boldsymbol{x}}^{(1)}, \tag{1}$$

where $\boldsymbol{b}$ is a bias vector, and $a$ is a non-linear activation function. It is important to note that for the output layer, the activation function is linear, and there is no bias vector.

The output layer in Eq. (1) involves only matrix multiplication, which is inherently a linear operation. Therefore, the decoder lacks the capacity to learn nonlinear mappings from a Euclidean space to a function space. The reconstruction error $\sum_{d=1}^p \|y^d - \hat{y}^d\|_{\mathcal{H}}^2$ is fundamentally constrained by how well the FD can be represented in a finite-dimensional linear subspace. When the FD lie on a low-dimensional submanifold, the FAE architecture requires a nonlinear decoder for efficient reconstruction. To incorporate nonlinearity, we extend the FAE architecture depicted in Figure 1 by introducing additional hidden layers, where each hidden layer is a composition of linear operations and a nonlinear activation function. For example, with two more hidden layers, the information flow is as follows:

$$\boldsymbol{y} \xrightarrow{\mathrm{w}_{q,d}^{(1)} \in \mathcal{H}(\mathcal{T}, \mathbb{R})} \boldsymbol{x}^{(1)} \xrightarrow{\mathrm{MLP}} \boldsymbol{x} \xrightarrow{\mathrm{MLP}} \hat{\boldsymbol{x}}^{(1)} \xrightarrow[\mathrm{Eq.(2)}]{\omega_{d,q}^{(1)} \in \mathcal{H}(\mathcal{T}, \mathbb{R})} \hat{\boldsymbol{y}}^{(1)} \xrightarrow[\mathrm{Eq.(3)}]{\omega_{d,q}^{(2)} \in \mathcal{H}(\mathcal{T}, \mathbb{R})} \hat{\boldsymbol{y}}^{(2)} \xrightarrow[\mathrm{Eq.(4)}]{\omega_{d,q}^{(3)} \in \mathcal{H}(\mathcal{T}, \mathbb{R})} \hat{\boldsymbol{y}}.$$

Note that the weights for the additional hidden layers are all continuous functions. The corresponding feedforward equations for the additional hidden layers and the output layer are:

$$\hat{\boldsymbol{y}}^{(1)}(t) \quad = \quad a(\boldsymbol{W}^{(1)}(t)\hat{\boldsymbol{x}}^{(1)} + \boldsymbol{b}_1(t)), \tag{2}$$

$$\hat{\boldsymbol{y}}^{(2)}(t) \quad = \quad a(\boldsymbol{W}^{(2)}(t)\hat{\boldsymbol{y}}^{(1)}(t) + \boldsymbol{b}_2(t)), \tag{3}$$

$$\hat{\boldsymbol{y}}(t) \quad = \quad \boldsymbol{W}^{(3)}(t)\hat{\boldsymbol{y}}^{(2)}(t), \tag{4}$$

where $\boldsymbol{b}_1(t)$ and $\boldsymbol{b}_2(t)$ are bias functions. Again, for the output layer, the activation function is linear, and there is no bias function.

While the encoder $\mathcal{E}$ is generally not an injective mapping, Stinchcombe [29] proved that $\mathcal{E}$ acts as a universal approximator when the activation function $a$ is continuous and non-polynomial. We establish below that the decoder structured as above is also a universal approximator in the Euclidean setting and achieves patchwise universal approximation when the output lies on a compact, connected Riemannian manifold. Theorem 1 ensures that the decoder can accurately reconstruct any latent representation, allowing the encoder to fully utilize the latent space without avoiding regions where the decoder might otherwise fail. The construction of the readout map $\rho$ is provided in Appendix B.2.

**Theorem 1.** *Let $\mathcal{F} = \{f : \mathbb{R}^s \mapsto \mathcal{H}(\mathcal{T}, \mathcal{M})\}$ denote a family of continuous mappings. When $\mathcal{M}$ is a $p$-dimensional Euclidean space, for any $f \in \mathcal{F}$ and $\epsilon \in (0, 1)$, there exists a functional network $\mathcal{D} : \mathbb{R}^s \mapsto \mathcal{H}(\mathcal{T}, \mathbb{R}^p)$ with the structure given by Eqs. (2-4) such that $\sup_{\boldsymbol{x} \in E} \|\mathcal{D}(\boldsymbol{x}) - f(\boldsymbol{x})\|_{\mathcal{H}} < \epsilon$ for any compact set $E \subset \mathbb{R}^s$. When $\mathcal{M}$ is a compact, connected Riemannian manifold that isometrically embeds in $\mathbb{R}^p$, for any $f \in \mathcal{F}$ and $\epsilon \in (0, 1)$, there exists a functional network $\mathcal{D} : \mathbb{R}^s \mapsto \mathcal{H}(\mathcal{T}, \mathbb{R}^p)$ with the structure given by Eqs. (2-4) and a readout map $\rho$, such that $\sup_{\boldsymbol{x} \in E} \|\rho \circ \mathcal{D}(\boldsymbol{x}) - f(\boldsymbol{x})\|_{\mathcal{H}} < \epsilon$ for controlled compact sets $E \subset \mathbb{R}^s$ whose maximal diameter depends on the curvature of $\mathcal{M}$.*

## 2.2   The joint training and clustering framework

Given the unsupervised nature of our problem, network training and cluster analysis must be performed jointly in each forward-backward loop. Specifically, during the forward phase, we update the learned latent representations $\{\mathcal{E}(\boldsymbol{y}_i)\}_{i=1}^n$, which necessitates a concurrent update of the clustering results. In the backward phase, we update the network parameters by minimizing a unified objective function that incorporates both the network training objective and the clustering regularizer.

Our methodological framework is depicted in Figure 2, where network update and cluster update are carried out jointly in an iterative manner. In the forward phase, clusters are updated by minimizing a clustering objective function $\mathcal{L}_s$; in the backward phase, network parameters are updated by minimizing an integrated objective function $\mathcal{L} = \mathcal{L}_r + \lambda_{\mathrm{w}}\mathcal{L}_{\mathrm{w}} + \lambda_c\mathcal{L}_c$. Here, $\mathcal{L}_r$ is the reconstruction loss, $\mathcal{L}_{\mathrm{w}}$ is a regularization term on the functional weights and biases, and $\mathcal{L}_c$ is a measure of clustering validity. The tuning parameter $\lambda_{\mathrm{w}}$ controls the amount of regular-

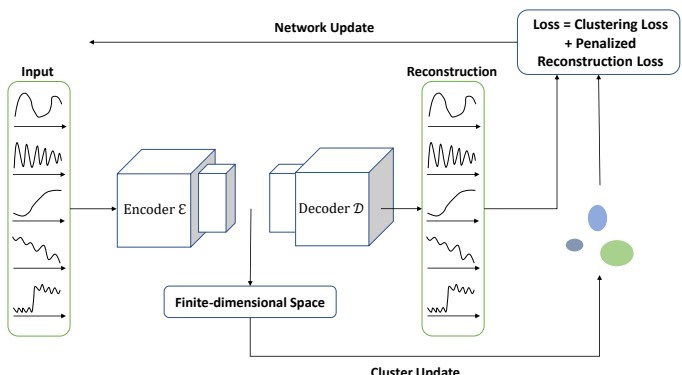

Figure 2: The joint network training and clustering framework.

ization. The combined term $\mathcal{L}_r + \lambda_{\mathrm{w}}\mathcal{L}_{\mathrm{w}}$ is interpreted as a penalized reconstruction loss. Incorporating the clustering loss $\mathcal{L}_c$ into the training objective encourages the encoder to learn representations that are conducive to effective clustering.

If an initial clustering of the FD is available, then training can be performed through backpropagation in the traditional way. Otherwise, if no initial clustering is available, we need to pre-train the network by minimizing only the penalized reconstruction loss $\mathcal{L}_r + \lambda_{\mathrm{w}}\mathcal{L}_{\mathrm{w}}$. After obtaining a partition of the embedded data, we then fine-tune the network by minimizing the integrated loss function $\mathcal{L}$.

# 3 Backward: network update

## 3.1 Clustering loss $\mathcal{L}_c$

To encourage the FAE to learn clustering-friendly representations, we need to incorporate a clustering-specific loss, denoted by $\mathcal{L}_c$, into the network training objective function. Let the generic notation $\boldsymbol{X} \in \mathbb{R}^{n \times s}$ denote the data matrix in the embedding space: $\boldsymbol{X}^T = [\boldsymbol{x}_1, \ldots, \boldsymbol{x}_n]$, and let $\{\mathcal{C}_k\}_{k=1}^K$ denote the current partition of the data $\boldsymbol{X}$, where $\mathcal{C}_k$ is the $k$th cluster with $n_k$ members. The fundamental concepts of clustering validity are compactness and separation. A natural measure of compactness is the within-cluster variance, and a natural measure of separation is the total distance from the cluster centroids to the overall sample average. To encourage both compactness within clusters and separation between clusters, we formulate the clustering loss penalty to be:

$$\mathcal{L}_c(\boldsymbol{X}) = \frac{1}{ns}\Big(\sum_{k=1}^K \sum_{\boldsymbol{x}_i \in \mathcal{C}_k} \|\boldsymbol{x}_i - \boldsymbol{\mu}_k\|_{\ell_2}^2 - \sum_{k=1}^K n_k \|\boldsymbol{\mu}_k - \bar{\boldsymbol{x}}\|_{\ell_2}^2\Big)$$

$$= \frac{1}{ns}\Big(2\sum_{k=1}^K \sum_{\boldsymbol{x}_i \in \mathcal{C}_k} \|\boldsymbol{x}_i - \boldsymbol{\mu}_k\|_{\ell_2}^2 - \sum_{i=1}^n \|\boldsymbol{x}_i - \bar{\boldsymbol{x}}\|_{\ell_2}^2\Big),$$

where $\boldsymbol{\mu}_k = \frac{1}{n_k} \sum_{\boldsymbol{x}_i \in \mathcal{C}_k} \boldsymbol{x}_i$ is the centroid of the cluster $\mathcal{C}_k$, and $\bar{\boldsymbol{x}} = \frac{1}{n} \sum_{1=1}^n \boldsymbol{x}_i$ is the overall sample average. The formulation of the clustering loss function is in accordance with the Calinski-Harabasz index, a clustering validation measure. The main difference is that we additionally include the dimension parameter $s$, to make the partition $\{\mathcal{C}_k\}_{k=1}^K$ comparable over different embedding spaces.

## 3.2 Penalized reconstruction loss $\mathcal{L}_r + \lambda_{\mathbf{w}} \mathcal{L}_{\mathbf{w}}$

The network parameters are weights and biases and are learned through minimizing a training objective function in the backpropagation manner. Let $\boldsymbol{\theta}$ denote the collection of all network parameters. The reconstruction loss of the $n$ input-output pairs $\{\boldsymbol{y}_i, \hat{\boldsymbol{y}}_i\}_{i=1}^n$ is

$$\mathcal{L}_r(\boldsymbol{\theta}) = \frac{1}{n} \sum_{i=1}^n \sum_{d=1}^p \|y_i^d - \hat{y}_i^d\|_{\mathcal{H}}^2.$$

In the relevant works reviewed before, the network parameters are learned by minimizing the reconstruction loss $\mathcal{L}_r(\boldsymbol{\theta})$ only. However, for each functional dimension $d$, without appropriately regularizing the functional weights $\{\mathrm{w}_{q,d}^{(1)} : 1 \leq q \leq q_1\}$, the training process might learn similar functional weights and hence extract redundant information from the functional input $y^d$. Likewise, the absence of constraints on the functional weights $\{\omega_{d,q}^{(1)} : 1 \leq q \leq \tilde{q}_1\}$ can render the training process unstable and prone to overfitting. We therefore introduce a few regularization terms into the reconstruction loss function to improve the stability, robustness, and generalization performance of the trained model.

**Orthogonality Penalty** To encourage different functional weights in the encoder to learn different (uncorrelated) information about the input function, we can regularize them to be orthogonal. Note that we only require the within-component functional weights $\{\mathrm{w}_{q,d}^{(1)} : 1 \leq q \leq q_1\}$ to be orthogonal, while the choice of functional weights for one component random function is independent of that for any other component random function. To encourage orthogonality, a natural choice of penalty is the $\ell_1$-type regularization:

$$\sum_{d=1}^p \Big[\sum_{1 \leq q < g \leq q_1} |\langle \mathrm{w}_{q,d}^{(1)}, \mathrm{w}_{g,d}^{(1)}\rangle_{\mathcal{H}}| + \sum_{q=1}^{q_1} |\|\mathrm{w}_{q,d}^{(1)}\|_{\mathcal{H}}^2 - 1|\Big].$$

However, the above functional is neither differentiable nor convex. In fact, even in vector calculus, an absolute inner product $|\langle \mathbf{w}_1, \mathbf{w}_2\rangle|$ is neither a differentiable nor a convex function of the argument pair $(\mathbf{w}_1, \mathbf{w}_2)$. Therefore, computing a functional variant of "subgradient" (to perform gradient descent) requires heavy machinery of the notion *variation* in the calculus of variations. We are not looking for exactly orthogonal functional weights, and hence we here adopt the $\ell_2$-type regularization (rather than the $\ell_1$-type regularization):

$$\mathcal{L}_{\mathrm{w}}(\boldsymbol{W}^{(1)}) = \sum_{d=1}^p \Big[\sum_{1 \leq q < g \leq q_1} \langle \mathrm{w}_{q,d}^{(1)}, \mathrm{w}_{g,d}^{(1)}\rangle_{\mathcal{H}}^2 + \sum_{q=1}^{q_1} (\|\mathrm{w}_{q,d}^{(1)}\|_{\mathcal{H}}^2 - 1)^2\Big].$$

To perform gradient backpropagation and directly optimize over the functional weights, a definition of functional derivative is required. A natural choice is the Fréchet derivative in the calculus of

variations [22, Chapter 4]. However, implementing optimization over a functional weight $\mathrm{w}_{q,d}^{(1)}$ in computer code entails optimizing the function's values over a fine grid, leading to significant computational overhead. Furthermore, while enforcing the orthogonal penalty $\mathcal{L}_{\mathrm{w}}(\boldsymbol{W}^{(1)})$ can aid in regularization, the optimized functional weights could exhibit violent fluctuations. A natural expectation for the functional weights $\{\mathrm{w}_{q,d}^{(1)} : 1 \leq q \leq q_1, 1 \leq d \leq p\}$ is their continuity and differentiability. Therefore, to uphold the continuity and differentiability of the functional weights $\{\mathrm{w}_{q,d}^{(1)} : 1 \leq q \leq q_1, 1 \leq d \leq p\}$ throughout the training process, we formulate the functional weights into linear combinations of predetermined (continuously differentiable) basis functions. This reformulation reduces the optimization problem from one over the functional space to one over the coefficient vectors of the basis expansion. This approach not only preserves the smoothness and differentiability of the functional weights but also significantly reduces computational complexity.

**Roughness Penalty** The functional weights (and functional biases) in the decoder are for approximating the FD, not for learning latent representations. Therefore, to prevent the functional weights and biases from exhibiting sharp bends or excessive curvature, we introduce a second-order roughness penalty, namely,

$$\mathcal{L}_{\mathrm{w}}(\boldsymbol{W}^{(l)}) = \sum\nolimits_{d,q} \int_{\mathcal{T}} (\omega_{d,q}^{(l)}{}''(t))^2 dt, \quad \mathcal{L}_{\mathrm{w}}(\boldsymbol{b}_l) = \sum\nolimits_{d} \int_{\mathcal{T}} (b_{l,d}''(t))^2 dt,$$

to penalize functional weights and functional biases with high curvature. Again, to maintain the continuity and differentiability of the functional weights and functional biases throughout the training process, we formulate them into linear combinations of predetermined (continuously differentiable) basis functions. Consequently, in the code implementation, the roughness penalty on, e.g., the functional weight $\omega_{d,q}^{(1)}$ is effectively replaced by the $\ell_1$ penalty on its basis-expansion coefficient vector. Here, we adopt the $\ell_1$ regularization to encourages sparsity in the coefficient vectors.

Incorporating all the regularization terms on the functional parameters, the penalized reconstruction loss function is given by:

$$\mathcal{L}_r(\boldsymbol{\theta}) + \lambda_{\mathrm{w}}[\mathcal{L}_{\mathrm{w}}(\boldsymbol{W}^{(1)}) + \sum\nolimits_{l} \mathcal{L}_{\mathrm{w}}(\boldsymbol{W}^{(l)}) + \sum\nolimits_{l} \mathcal{L}_{\mathrm{w}}(\boldsymbol{b}_l)],$$

where the penalty terms on the weights and biases are functions of their basis-expansion coefficient vectors. In other words, by representing the functional weights and functional biases as linear expansions of basis functions, all the trainable parameters in the FAE become scalar values, allowing us to apply standard scalar backpropagation techniques.

The scalar weights in the two MLPs are regularized using batch normalization and dropout. Details of these techniques, as implemented in our Python framework `FAEclust`, are provided in Appendix C. The optimization algorithm used to train the FAE is described in Appendix D.

## 4 Forward: cluster update

### 4.1 Clustering objective function

Once the latent representations are available, we then apply a clustering method on the embedded data to obtain clusters. We want the clustering method to be able to adapt to the original FD, rather than completely ignoring the original FD. One appropriate choice is the technique of convex clustering [36], with the clustering objective function being:

$$\mathcal{L}_s(\boldsymbol{U}, \boldsymbol{X}) = \frac{1}{n}\|\boldsymbol{X} - \boldsymbol{U}\|_F^2 + \lambda \sum\nolimits_{1 \leq i < j \leq n} \mathbb{s}(\boldsymbol{y}_i, \boldsymbol{y}_j)\|\boldsymbol{u}_i - \boldsymbol{u}_j\|_{\ell_1}, \tag{5}$$

where $\|\cdot\|_F$ is the Frobenius norm, and $\mathbb{s}(\boldsymbol{y}_i, \boldsymbol{y}_j)$ is a similarity/affinity measure between the sample functions $\boldsymbol{y}_i$ and $\boldsymbol{y}_j$. Here, $\boldsymbol{U}^T = [\boldsymbol{u}_1, \ldots, \boldsymbol{u}_n]$, and $\boldsymbol{u}_i \in \mathbb{R}^s$ $(i = 1, \ldots, n)$ is the centroid of the cluster that object $\boldsymbol{x}_i$ belongs to. Therefore, by forcing $\boldsymbol{u}_i = \boldsymbol{u}_j$, the two data points $\boldsymbol{x}_i$ and $\boldsymbol{x}_j$ will belong to the same cluster. The second term in Eq. (5) is a regularizer, putting a constraint on the number of distinct cluster centroids. If $\lambda = 0$, the minimum is attained when $\boldsymbol{U} = \boldsymbol{X}$, and each point $\boldsymbol{x}_i$ occupies a unique cluster $\boldsymbol{u}_i$. As $\lambda$ increases, the cluster centroids begin to coalesce. For sufficiently large $\lambda$, all related points will coalesce into a single cluster. In Section 4.2, we develop an efficient algorithm for minimizing the loss function $\mathcal{L}_s(\boldsymbol{U}, \boldsymbol{X})$ w.r.t. $\boldsymbol{U}$, for any value of $\lambda (> 0)$.

For traditional multivariate data, the similarity measure $\mathbb{s}(\cdot,\cdot)$ is usually defined to be distance-dependent, e.g. $\mathbb{s}(\boldsymbol{x}_i,\boldsymbol{x}_j) = \exp(-\|\boldsymbol{x}_i - \boldsymbol{x}_j\|_{\ell_2})$, in order to make the estimates of the centroids enjoy asymptotic consistency. Let $\mathrm{d}(\cdot,\cdot)$ denote a distance metric, and $\mathcal{N}_m(\boldsymbol{y}_i)$ denote the set of $m$-nearest "neighbors" of $\boldsymbol{y}_i$, where the neighbors are determined by the distance metric $\mathrm{d}(\cdot,\cdot)$. In analogy to [36], the similarity measure $\mathbb{s}(\boldsymbol{y}_i,\boldsymbol{y}_j)$ can be formulated as

$$\mathbb{s}(\boldsymbol{y}_i,\boldsymbol{y}_j) = \delta(\boldsymbol{y}_j \in \mathcal{N}_m(\boldsymbol{y}_i) \text{ or } \boldsymbol{y}_i \in \mathcal{N}_m(\boldsymbol{y}_j)) \times \exp(-\mathrm{d}(\boldsymbol{y}_i,\boldsymbol{y}_j)).$$

The introduction of the neighbor set $\mathcal{N}_m(\boldsymbol{y}_i)$ aims to preserve the locality of the clusters by pushing nearby data points together. In our Python framework `FAEclust`, we offer two options for determining the optimal neighborhood size $m$: the distance knee method and graph connectivity analysis.

Unlike traditional multivariate data, FD may differ in two types of variation: amplitude variation in curve height and phase variation in lateral displacements of curve features (e.g. peaks, points of inflection, and threshold crossings). If no phase variation is presented in the sample functions, or if it is the joint variation between amplitude and phase that determines the clusters, then $\mathrm{d}$ is the standard distance metric for Hilbert spaces: $\mathrm{d}(\boldsymbol{y}_i,\boldsymbol{y}_j) = \sqrt{\sum_{d=1}^{p} \|y_i^d - y_j^d\|_{\mathcal{H}}^2}$. If amplitude variation is the main focus, with phase variation being a nuisance, then we can utilize the square-root velocity (SRV) framework [27]. In particular, we require that any sample function $\boldsymbol{y}_i$ is absolutely continuous on $\mathcal{T}$; that is, the component functions $\{y_i^d\}_{d=1}^{p}$ are all absolutely continuous. Let $H = \{h : \mathcal{T} \to \mathcal{T}\}$ denote the set of all orientation-preserving diffeomorphisms. Then for any $h \in H$, the composition $\boldsymbol{y}_i \circ h$ is a re-parameterization of $\boldsymbol{y}_i$. The SRV representation of $\boldsymbol{y}_i$ is

$$\mathrm{SRV}(\boldsymbol{y}_i)(t) = \begin{cases} \boldsymbol{y}_i'(t)/\sqrt{\|\boldsymbol{y}_i'(t)\|_{\ell_2}}, & \|\boldsymbol{y}_i'(t)\|_{\ell_2} \neq 0; \\ 0, & \text{otherwise.} \end{cases}$$

Then the SRV representation of the re-parameterized function $\boldsymbol{y}_i \circ h$ is

$$\mathrm{SRV}(\boldsymbol{y}_i \circ h)(t) = \mathrm{SRV}(\boldsymbol{y}_i)(h(t))\sqrt{h'(t)}.$$

The distance between $\boldsymbol{y}_i$ and $\boldsymbol{y}_j$ is defined to be

$$\mathrm{d}(\boldsymbol{y}_i,\boldsymbol{y}_j) = \inf_{h \in H} \mathrm{d}_{FR}(\boldsymbol{y}_i \circ h, \boldsymbol{y}_j) = \inf_{h \in H} \|\mathrm{SRV}(\boldsymbol{y}_i \circ h) - \mathrm{SRV}(\boldsymbol{y}_j)\|_{\mathcal{H}^p}, \tag{6}$$

where $\mathrm{d}_{FR}$ is the Fisher-Rao Riemannian distance metric.

The distance metric defined in Eq. (6) has the property that it is invariant to both translation and reparameterization. Its computation relies on dynamic programming, which, however, has a time complexity of $\mathcal{O}(N^2)$, where $N$ is the number of nodes of a grid on the interval $\mathcal{T}$ [26, Appendix B]. Therefore, calculating the pairwise distances $\{\mathrm{d}(\boldsymbol{y}_i,\boldsymbol{y}_j) : 1 \leq i < j \leq n\}$ requires substantial computational resources for datasets of medium to large size. To address this, our Python framework `FAEclust` additionally implements fast and ultra-fast dynamic time warping (DTW) techniques to approximate the similarity $\mathbb{s}(\boldsymbol{y}_i,\boldsymbol{y}_j)$. The fast DTW algorithm achieves a time complexity of $\mathcal{O}(N)$, while the ultra-fast DTW [21] further reduces complexity to $\sim\mathcal{O}(N)$, enabling scalable and rapid computation for large datasets.

## 4.2 Optimization algorithm

The objective function $\mathcal{L}_s(\boldsymbol{U},\boldsymbol{X})$ is separable on dimensions, and hence the minimization can be carried out separately in parallel for each embedding dimension. We might let the generic vector $\boldsymbol{x}$ represent an arbitrary column of $\boldsymbol{X}$, and $\boldsymbol{u}$ the corresponding column of $\boldsymbol{U}$. Minimizing $\mathcal{L}_s(\boldsymbol{U},\boldsymbol{X})$ amounts to solving $s$ minimization problems of the following form: $\boldsymbol{u}(\lambda) = \arg\min_{\boldsymbol{u} \in \mathbb{R}^n} \mathcal{L}_1(\boldsymbol{u})$, where

$$\mathcal{L}_1(\boldsymbol{u}) = \frac{1}{n}\|\boldsymbol{x} - \boldsymbol{u}\|_{\ell_2}^2 + \lambda \sum_{1 \leq i < j \leq n} \mathbb{s}(\boldsymbol{y}_i,\boldsymbol{y}_j)|u_i - u_j|.$$

We here develop a path-following homotopy algorithm with a complexity of $O(n\log(n))$ for finding the solution $\boldsymbol{u}(\lambda)$ for any value of $\lambda > 0$, utilizing the property that each element $u_i(\lambda)$ is a piecewise linear function of the parameter $\lambda$ [9].

According to Eq. (5), for a given embedding $\boldsymbol{X}$, the convex clustering objective admits a unique solution for any value of $\lambda$. When $\lambda = 0$, the solution is simply $\boldsymbol{u}(0) = \boldsymbol{x}$, assigning each point to its own cluster. As $\lambda$ increases, pairs of cluster centroids merge sequentially, giving rise to a complete

agglomerative clustering hierarchy. Crucially, these merges happen only at specific values of $\lambda$, referred to as breakpoints. Let $\lambda$ take the value $\lambda_K$, the breakpoint at which two clusters are being merged into one, and now the data are partitioned into $K$ clusters; that is, there are $K$ unique values in the solution $\boldsymbol{u}(\lambda_K)$, denoted by $\{\ddot{u}_1, \ldots, \ddot{u}_K\}$. Right after the merge, the algorithm computes the next breakpoint $\lambda_{K-1}$, at which the next merge will happen, producing a partition with $(K-1)$ clusters. When $\lambda$ lies between the two consecutive breakpoints, namely $\lambda_K \leq \lambda < \lambda_{K-1}$, the clustering remains unchanged. Our path-following homotopy algorithm efficiently identifies all the breakpoints in $\mathcal{O}(n \log(n))$ time, thereby constructing the full clustering hierarchy. The computational details of this procedure are presented in Appendix E. A high-level outline of the algorithm is provided below.

**Input**: latent representations $\boldsymbol{X}$ and similarity measures $\{\mathbb{s}(\boldsymbol{y}_i, \boldsymbol{y}_j) : 1 \leq i < j \leq n\}$.
**while** number of clusters $> 2$ **do**
    Compute the next breakpoint $\lambda_K$ where a pair of centroids merge.
    Update the clustering by merging the corresponding pair of clusters.
**end while**
**Output**: breakpoints $\{\lambda_{n-1}, \ldots, \lambda_2\}$ and corresponding hierarchical clustering.

Once the hierarchy is constructed, an internal validation index (such as the silhouette score or Davies-Bouldin index) is used to select the optimal clustering result, thereby determining the number of clusters in a data-driven manner. This selection is entirely independent of any particular $\lambda$ value; that is, the $\lambda$ parameter in Eq. (5) functions as an algorithmic variable for path tracing, not as a tunable hyperparameter.

The network training process consists of several iterations of the forward-backward loop. In each forward pass, we repeat the hierarchy-construction and clustering-selection procedure described above. The final clustering result from the last iteration is reported as the output of the model.

## 5 Experiments

We selected four FD clustering algorithms available on CRAN – `funHDDC`, `funclust`, `FADPclust` (`FADP1 & FADP2`) – for benchmarking because they natively support multi-dimensional functional datasets, where the input is a list of `fd` objects (one per dimension). Other FD clustering methods available on CRAN either do not accept `fd` objects as input (e.g., `kmeans_align`) or reduce multi-dimensional functions to a one-dimensional form, e.g., by concatenating the basis-expansion coefficients. On the Python side, we include the VANO model [25], whose reference implementation is available on GitHub. Additionally, we re-implemented the methods from [8] (FNN) and [10] (FAE) ourselves as no official implementations are currently available. To ensure consistency, our FNN uses the architecture depicted in Figure 1 of [8], and our FAE follows Algorithm 1 of [10]. In Appendix F.1, we detail the input arguments of our `FAEclust` algorithm and summarize the configuration settings used for every algorithm in this benchmark.

When the FD are in a Euclidean space, we applied all eight FD clustering algorithms to nine one-dimensional and eight multi-dimensional functional datasets from the UEA & UCR Time Series Classification Repository. Clustering performance, evaluated using the Adjusted Mutual Information (AMI), is summarized in Table 1. Results on the Adjusted Rand Index (ARI) are provided in Appendix F.3. Dataset abbreviations used in the table include: DSR (DiatomSizeReduction), BM (BasicMotions), SWJ (StandWalkJump), EOS (EyesOpenShut), FM (FingerMovements) and JV (JapaneseVowels). `FAEclust` is the most consistent performer: it achieves the best AMI on 12 of 17 datasets and ranks top-2 on 15 of 17. Table 1 highlights the robustness and generalizability of `FAEclust` in clustering both univariate and multivariate FD.

For manifold-valued FD, we investigate five types of manifolds: Hypersphere, Hyperbolic, Swiss roll, Lorenz, and Pendulum. Details on the simulated FD are given in Appendix F.2. For each simulation scenario, we apply each model on 100 simulated datasets and report the mean and standard deviation of AMI (Table 2) and ARI (Table 7). Figure 10 in Appendix F.3 shows box plots of the number of clusters identified across the 100 repetitions. `FAEclust` attains the best performance on all five manifolds, with especially strong gains on nonlinear dynamics (e.g., Hyperbolic and Lorenz) and near-perfect accuracy on Pendulum, while exhibiting low variability across repetitions.

We evaluate the robustness of `FAEclust` to phase variation by extending the 12 simulation scenarios from [1]. For each scenario, we begin by generating an unwarped functional dataset and applying `FAEclust`. We then introduce phase variation by composing each function with a randomly gener-

Table 1: AMI scores for the 17 Euclidean functional datasets.

| Dataset | funHDDC | funclust | FADP1 | FADP2 | FNN | FAE | VANO | FAEclust |
|---|---|---|---|---|---|---|---|---|
| BirdChicken | 0.055 | 0.080 | 0.019 | 0.055 | 0.290 | 0.259 | 0.302 | **0.339** |
| CBF | 0.362 | 0.066 | 0.435 | 0.318 | 0.363 | 0.370 | **0.743** | 0.724 |
| Chinatown | 0.147 | 0.147 | 0.246 | 0.055 | 0.236 | 0.110 | 0.339 | **0.343** |
| DSR | 0.786 | 0.017 | 0.692 | 0.876 | 0.679 | 0.645 | 0.713 | **0.887** |
| ECG200 | 0.173 | 0.101 | 0.201 | 0.143 | 0.214 | 0.213 | 0.217 | **0.261** |
| Fungi | 0.773 | 0.186 | 0.477 | 0.501 | 0.244 | 0.624 | 0.798 | **0.925** |
| Plane | 0.819 | 0.013 | 0.725 | 0.741 | 0.841 | 0.825 | 0.846 | **0.907** |
| Rock | 0.373 | 0.228 | 0.216 | 0.335 | 0.184 | 0.089 | 0.355 | **0.447** |
| Symbols | 0.767 | 0.001 | 0.435 | 0.712 | 0.748 | 0.800 | 0.817 | **0.824** |
| Blink | 0.410 | 0.048 | 0.189 | 0.177 | 0.453 | 0.506 | 0.522 | **0.633** |
| BM | 0.377 | 0.031 | 0.191 | 0.422 | 0.401 | **0.676** | 0.592 | 0.539 |
| EOS | 0.209 | 0.018 | 0.104 | 0.102 | 0.152 | 0.206 | 0.180 | **0.266** |
| Epilepsy | 0.143 | 0.028 | 0.077 | 0.225 | 0.274 | 0.209 | 0.297 | **0.485** |
| ERing | 0.735 | 0.012 | 0.288 | 0.714 | 0.643 | **0.743** | 0.733 | 0.664 |
| FM | 0.001 | 0.001 | 0.002 | 0.002 | 0.174 | 0.138 | 0.174 | **0.228** |
| JV | 0.840 | 0.069 | 0.294 | 0.466 | 0.236 | 0.854 | **0.899** | 0.893 |
| SWJ | 0.268 | 0.040 | 0.248 | 0.046 | 0.174 | 0.170 | **0.344** | 0.324 |

Table 2: AMI scores for the five manifold-valued functional datasets. The table reports the mean (top row) and standard deviation (bottom row) of the scores over 100 repetitions.

| Dataset | funHDDC | funclust | FADP1 | FADP2 | FNN | FAE | VANO | FAEclust |
|---|---|---|---|---|---|---|---|---|
| Hypersphere | 0.016 | 0.478 | 0.137 | 0.067 | 0.089 | 0.307 | 0.443 | **0.737** |
| | 0.041 | 0.036 | 0.127 | 0.071 | 0.038 | 0.052 | 0.063 | **0.026** |
| Hyperbolic | 0.005 | 0.013 | 0.001 | 0.001 | 0.004 | 0.047 | 0.410 | **0.798** |
| | 0.013 | 0.044 | 0.006 | 0.006 | **0.003** | 0.016 | 0.042 | 0.034 |
| Swiss roll | 0.127 | 0.114 | 0.382 | 0.189 | 0.016 | 0.125 | 0.242 | **0.432** |
| | 0.059 | 0.035 | 0.046 | 0.065 | **0.026** | 0.035 | 0.062 | 0.040 |
| Lorenz | 0.109 | 0.389 | 0.092 | 0.144 | 0.023 | 0.246 | 0.251 | **0.457** |
| | 0.057 | 0.049 | 0.060 | 0.081 | 0.047 | **0.022** | 0.034 | 0.038 |
| Pendulum | 0.887 | 0.376 | 0.797 | 0.808 | 0.253 | 0.794 | 0.905 | **0.986** |
| | 0.029 | 0.058 | 0.049 | 0.063 | 0.044 | 0.074 | 0.038 | **0.006** |

ated time-warping function, created using the `tsaug._augmenter.time_warp` module in Python. `FAEclust` is subsequently reapplied to the time-warped dataset. This entire procedure is repeated 100 times, and Table 3 reports the mean and standard deviation of ARI and AMI scores across 100 repetitions for both the original (unwarped) and time-warped datasets. Figure 13 in Appendix F.4 shows box plots of the number of clusters identified across the 100 repetitions. Tables 8 and 9 in Appendix F.4 summarize the AMI and ARI performance, respectively, of the seven baseline methods on the time-warped datasets.

The results in Tables 3, 8 and 9 show that `FAEclust` exhibits strong robustness to phase variation: introducing random time warping changes the mean AMI/ARI by only 0.013 on average. Performance remains essentially unchanged and near-perfect in structured settings, while the largest degradations are modest. Against seven baselines on the warped datasets, `FAEclust` attains the highest scores in 11/12 scenarios and does so by substantial margins; for example, relative to the best baseline it improves by +0.10/+0.199 (AMI/ARI) in A, +0.315/+0.311 in B, +0.093/+0.121 in E, and +0.501/+0.467 in K. The only exception is scenario L, where FNN attains a slightly higher AMI (0.728 vs. 0.699), but `FAEclust` still yields the best ARI (0.709 vs. 0.667). Overall, these results demonstrate that `FAEclust` is highly robust to phase variation and provides state-of-the-art clustering accuracy on time-warped functional data.

Table 3: AMI and ARI scores of `FAEclust` for the 12 simulation scenarios. For both the original (unwarped) and time-warped datasets, the table reports the mean (top row) and standard deviation (bottom row) of the scores over 100 repetitions.

| Scenario | Original | | Warped | | Scenario | Original | | Warped | |
|---|---|---|---|---|---|---|---|---|---|
| | AMI | ARI | AMI | ARI | | AMI | ARI | AMI | ARI |
| A | 0.782 | 0.730 | 0.754 | 0.723 | G | 0.582 | 0.484 | 0.580 | 0.492 |
| | 0.026 | 0.024 | 0.020 | 0.022 | | 0.106 | 0.110 | 0.098 | 0.116 |
| B | 0.428 | 0.441 | 0.412 | 0.385 | H | 1.000 | 1.000 | 1.000 | 1.000 |
| | 0.060 | 0.085 | 0.074 | 0.092 | | 0.000 | 0.000 | 0.000 | 0.000 |
| C | 0.820 | 0.751 | 0.782 | 0.710 | I | 0.997 | 0.993 | 0.996 | 0.992 |
| | 0.042 | 0.057 | 0.036 | 0.045 | | 0.007 | 0.004 | 0.009 | 0.006 |
| D | 0.581 | 0.514 | 0.556 | 0.518 | J | 0.149 | 0.086 | 0.145 | 0.110 |
| | 0.085 | 0.069 | 0.072 | 0.058 | | 0.065 | 0.061 | 0.074 | 0.066 |
| E | 0.985 | 0.990 | 0.983 | 0.990 | K | 0.871 | 0.819 | 0.867 | 0.813 |
| | 0.016 | 0.010 | 0.019 | 0.012 | | 0.030 | 0.026 | 0.036 | 0.030 |
| F | 0.888 | 0.872 | 0.869 | 0.819 | L | 0.712 | 0.734 | 0.699 | 0.709 |
| | 0.031 | 0.050 | 0.057 | 0.079 | | 0.049 | 0.067 | 0.056 | 0.081 |

## 6 Conclusion

We introduced `FAEclust`, a novel deep functional autoencoder framework for clustering multi-dimensional functional data. By combining universal-approximation guarantees for both encoder and decoder with a shape-aware clustering objective, `FAEclust` remains robust to phase variation and can capture complex, nonlinear (and even non-Euclidean) data geometries. We operationalize clustering through a convex objective and derive a path-following homotopy algorithm that constructs the full clustering hierarchy in $\mathcal{O}(n \log(n))$, with model selection performed along the path via internal validation. Extensive benchmarking on real-world and simulated datasets demonstrated the clear advantages of `FAEclust` over existing methods. Overall, `FAEclust` sets a new standard for clustering complex functional data by integrating geometric structure, regularized functional representations, and a shape-aware clustering loss into a unified deep learning framework.

## Acknowledgments

This work was conducted with the financial support of the Research Ireland Centre for Research Training in Digitally-Enhanced Reality (d-real) under Grant No. 18/CRT/6224. We are grateful to the four anonymous reviewers for their valuable suggestions.

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

# A Preliminaries

While the $n$ sample functions $\{\boldsymbol{y}_i : i = 1, \ldots, n\}$ are called *functional data*, in real applications, we only have access to their discrete evaluations. Moreover, the observation of $\boldsymbol{y}_i$ at any time point $t$ may come with an additive error. Hence, we write $\tilde{\boldsymbol{y}}_i(t) = \boldsymbol{y}_i(t) + \boldsymbol{\epsilon}_i(t)$, where $\epsilon_i^d(\cdot)$ is the noise process with $\mathrm{E}[\epsilon_i^d(t)] = 0$, $\mathrm{E}[\epsilon_i^d(t)^2] = \sigma_d^2$, and $\mathrm{cov}(\epsilon_i^d(t), \epsilon_i^v(s)) = 0$ for $s \neq t$ and $1 \leq d, v \leq p$. The sampling scheme for the $i$th subject is denoted by the column vector $\boldsymbol{t}_i = (t_{i1}, \ldots, t_{ir_i})^T$, where $t_{ij} \in \mathcal{T}$ for $j = 1, \ldots, r_i$, and $t_{i1} < \cdots < t_{ir_i}$. Here, to avoid overly complex notation, we assume that all component random functions have an identical function domain $\mathcal{T}$, and the $p$ component functions $\{y_i^1, \ldots, y_i^p\}$ of each subject have the same sampling scheme. The sequence of observations $\{\tilde{\boldsymbol{y}}_i(t_{i1}), \ldots, \tilde{\boldsymbol{y}}_i(t_{ir_i})\}$ is called a *sample path* of the sample function $\boldsymbol{y}_i$. The sample path of a one-dimensional sample function $y_i^d$ is written as $\tilde{y}_i^d(\boldsymbol{t}_i)$, where we employ compact notation for functions applied to collections of input points.

If a sample function $\boldsymbol{y}_i$ belongs to the $k$th cluster ($1 \leq k \leq K$), then it is a realization of the $k$th random function $\vec{Y}_k$, defined on the probability space $(\Omega, \mathcal{F}, \mathrm{Pr}_k)$. In other words, there are $K$ different probability measures defined on the $\sigma$-algebra $(\Omega, \mathcal{F})$. Given the sample paths $\{\tilde{\boldsymbol{y}}_i(t_{i1}), \ldots, \tilde{\boldsymbol{y}}_i(t_{ir_i})\}_{i=1}^n$, a clustering method is to identify the underlying random function for each sample function $\boldsymbol{y}_i$. Table 4 summarizes the notations we will use throughout the work. All vectors

Table 4: Notations adopted throughout the paper.

| | |
|---|---|
| $n$ | number of subjects |
| $i$ | index for subject, $1 \leq i \leq n$ |
| $p$ | number of component random functions |
| $d$ | index for dimension, $1 \leq d \leq p$ |
| $K$ | number of clusters in the population |
| $k$ | index for cluster, $1 \leq k \leq K$ |
| $\mathcal{C}_k$ | $k$th cluster of size $n_k$: $n = \sum_{k=1}^K n_k$ |
| $\{\boldsymbol{y}_i : i = 1, \ldots, n\}$ | $n$ independent $p$-dimensional sample functions |
| $\mathcal{T}$ | compact function domain $t \in \mathcal{T} \subset \mathbb{R}$ |
| $\epsilon_i^d(t)$ | white noise with variance $\sigma_d^2$ for $d = 1, \ldots, p$ |
| $\tilde{\boldsymbol{y}}_i(t)$ | observation of $\boldsymbol{y}_i(t)$: $\tilde{\boldsymbol{y}}_i(t) = \boldsymbol{y}_i(t) + \boldsymbol{\epsilon}_i(t)$ |
| $\boldsymbol{t}_i = (t_{i1}, \ldots, t_{ir_i})^T$ | sampling scheme for the $i$th subject |
| $r_i$ | number of sampling points in the $i$th sampling scheme |
| $\{\tilde{\boldsymbol{y}}_i(t_{i1}), \ldots, \tilde{\boldsymbol{y}}_i(t_{ir_i})\}$ | sample path of the $p$-dimensional sample function $\boldsymbol{y}_i$ |
| $\tilde{y}_i^d(\boldsymbol{t}_i)$ | sample path of the component sample function $y_i^d$ |
| $\vec{Y}_k$ | $k$th random function $\vec{Y}_k = (Y_k^1(t, \omega), \ldots, Y_k^p(t, \omega))^T$ |
| $(\Omega, \mathcal{F}, \mathrm{Pr}_k)$ | probability space for $\vec{Y}_k$, $k = 1, \ldots, K$ |
| $\langle \cdot, \cdot \rangle_{\mathcal{H}}, \|\cdot\|_{\mathcal{H}}$ | inner product and norm for the Hilbert space $\mathcal{H}(\mathcal{T}, \mathbb{R}^p)$ |
| $\|\cdot\|_{\ell_1}, \|\cdot\|_{\ell_2}$ | $\ell_1$ and $\ell_2$ norm on an Euclidean space |
| $\boldsymbol{y}'(t)$ | first-order derivative of the function $\boldsymbol{y}(t)$ |
| $\delta(\cdot)$ | 0-1 indicator function |
| sgn | signum function |

are column vectors.

The network model developed in Section 2 only accepts as input continuous functions $\{\boldsymbol{y}_i : i = 1, \ldots, n\}$. In general, however, we do not have access to any sample function $\boldsymbol{y}_i$, but only its noise-contaminated discrete evaluations $\{\tilde{\boldsymbol{y}}_i(t_{i1}), \ldots, \tilde{\boldsymbol{y}}_i(t_{ir_i})\}$. Therefore, in practice, we need to perform smoothing before cluster analysis.

**Smoothing**   Given the sample path $\{\tilde{\boldsymbol{y}}_i(t_{i1}), \ldots, \tilde{\boldsymbol{y}}_i(t_{ir_i})\}$, we need to recover the underlying continuous function $\boldsymbol{y}_i(t)$, which is commonly achieved through the smoothing technique. Let $\{b_1, \ldots, b_m\}$ be $m$ element functions of a pre-determined basis (e.g., the cubic B-spline basis) defined on the interval $\mathcal{T}$. Then we can approximate each component sample function $y_i^d$ by $\sum_{v=1}^m c_{iv}^d b_v$, where the coefficient vector $\boldsymbol{c}_i^d = (c_{i1}^d, \ldots, c_{im}^d)^T$ is commonly obtained through minimizing a

regularized residual sum of squares:

$$\min_{\boldsymbol{c}_i^d \in \mathbb{R}^m} \{\sum_{r=1}^{r_i} [\tilde{y}_i^d(t_{ir}) - \sum_{v=1}^{m} c_{iv}^d b_v(t_{ir})]^2 + \text{regularizer}\}.$$

The choice of the smoothing technique depends on fields of application: wavelets are effective in capturing discrete jumps or edges and especially useful for modeling signals and images; splines are more often applied for approximating a function of a given order. Kernel smoothing is another common method of estimating the mean function in the regression model $\tilde{y}_i^d(t) = y_i^d(t) + \epsilon_i^d(t)$. From Section 2, we formulate our model in terms of the sample functions $\{\boldsymbol{y}_i : i = 1, \ldots, n\}$ only, with the implication that they will be replaced by their smooth estimates in real applications.

**Standardization** As with the analysis of traditional multivariate data, where a pre-processing step is to standardize the data to have unit standard deviation in each dimension, Chiou et al. [5] proposed to standardize each component sample function to account for differences in degrees of variability and in units of measurements among the component random functions. The formula is in analogy to standardizing multivariate data: each component sample function is subtracted by its mean function and then divided by the square root of the variance function. Standardization is necessary if the component random functions have quite different ranges or if they exhibit different amounts of variation. If the sample functions are not scaled, algorithms that compute the overall distance $\sqrt{\sum_{d=1}^{p} \|y_i^d - y_j^d\|_{\mathcal{H}}^2}$ are biased towards component random functions with numerically larger values. Therefore, standardization is a crucial step for distance-based clustering algorithms. Moreover, standardization helps machine learning algorithms train and converge faster. Henceforth, we assume that standardization has been done before applying the network model.

# B Proof of Theorem 1

## B.1 Euclidean Universal Approximation

Existing universal approximation theorems primarily focus on mappings between finite-dimensional topological vector spaces. While some works have explored the universal approximation property for nonlinear functionals [29] and nonlinear operators [4], only a single study has established universality for mappings from a finite-dimensional space to an infinite-dimensional space [7]. Unlike previous approaches, which rely on integral kernels, our network architecture employs multiplicative parameters defined by continuous univariate functions. This key distinction necessitates a tailored universality proof for our specific framework, which we present here.

When $\mathcal{M}$ is the $p$-dimensional Euclidean space $\mathbb{R}^p$, it suffices to only consider the case $p = 1$. This is because uniform convergence in $\mathcal{H}(\mathcal{T}, \mathbb{R}^p)$ reduces to uniform convergence in each component space $\mathcal{H}(\mathcal{T}, \mathbb{R})$. We now prove that a neural network $\mathcal{D}$ with a single hidden layer can uniformly approximate any continuous mapping $f : \mathbb{R}^s \mapsto \mathcal{H}(\mathcal{T}, \mathbb{R})$ over any compact domain $E \subset \mathbb{R}^s$; see Figure 3. Given that $f$ is a continuous mapping from a compact set $E \subset \mathbb{R}^s$ into $\mathcal{H}(\mathcal{T}, \mathbb{R})$, it is straightforward to prove that the range $f(E) = \{f(\boldsymbol{x}) : \boldsymbol{x} \in E\}$ is a compact set in $\mathcal{H}(\mathcal{T}, \mathbb{R})$.

With the notation given in Figure 3, define $\boldsymbol{w}_{q.}^{(1)}(t) = (w_{q,1}^{(1)}(t), \ldots, w_{q,s}^{(1)}(t))^T$, where $m$ is the number of nodes in the hidden layer. The output function is

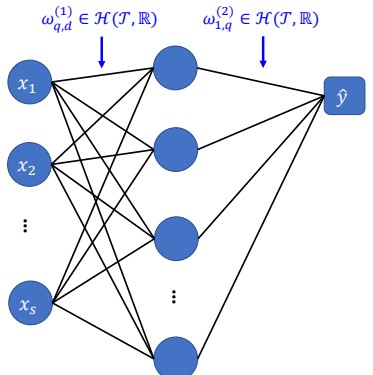

$$\mathcal{D}(\boldsymbol{x})(t) = \sum_{q=1}^{m} w_{1,q}^{(2)}(t) a(\boldsymbol{w}_{q.}^{(1)}(t)^T \boldsymbol{x} + b_q(t))$$

$$= \boldsymbol{w}^{(2)}(t)^T a(\boldsymbol{\mathcal{W}}^{(1)}(t)\boldsymbol{x} + \boldsymbol{b}(t)), \qquad (7)$$

Figure 3: A functional network $\mathcal{D} : \mathbb{R}^s \mapsto \mathcal{H}(\mathcal{T}, \mathbb{R})$ with only one hidden layer.

where $\boldsymbol{w}^{(2)}(t) = (w_{1,1}^{(2)}(t), \ldots, w_{1,m}^{(2)}(t))^T$ and $\boldsymbol{\mathcal{W}}^{(1)}(t)^T = [\boldsymbol{w}_{1.}^{(1)}(t), \ldots, \boldsymbol{w}_{m.}^{(1)}(t)]$. The activation function $a$ is chosen as a Tauber-Wiener function (e.g., sigmoid) [4]; that is, the set of all linear

combinations $\sum_{q=1}^{m} c_q a(w_q x + b_q)$ over a closed interval $I \subset \mathbb{R}$ is dense in $\mathbb{C}(I)$, where $c_q$, $w_q$, and $b_q$ are real numbers. The feedforward equation (7) differs from that of the traditional three-layer feedforward network only in that the parameters are now functions, varying continuously with the value of the output function. One can immediately claim that, for any continuous mapping $f : \mathbb{R}^s \mapsto \mathcal{H}(\mathcal{T}, \mathbb{R})$ and any fixed $t \in \mathcal{T}$, there exist (constant) weight parameters $\{w^{(2)}(t), \mathcal{W}^{(1)}(t), b(t)\}$ such that $\sup_{\boldsymbol{x} \in E} |\mathcal{D}(\boldsymbol{x})(t) - f(\boldsymbol{x})(t)| < \epsilon$ for any compact set $E \subset \mathbb{R}^s$; see, e.g., [15]. This leads to the following property:

**Property 1.** *Let $\mathbb{C}^0(\mathcal{T}, \mathbb{R}) \subset \mathcal{H}(\mathcal{T}, \mathbb{R})$ be the set of all piecewise constant functions over the compact interval $\mathcal{T}$. For any continuous mapping $f_c : \mathbb{R}^s \mapsto \mathbb{C}^0(\mathcal{T}, \mathbb{R})$, there exist piecewise constant weight parameters $\{w^{(2)}(t), \mathcal{W}^{(1)}(t), b(t)\}$ such that $\sup_{\boldsymbol{x} \in E} \|\mathcal{D}(\boldsymbol{x}) - f_c(\boldsymbol{x})\|_\mathcal{H} < \epsilon$ for any compact set $E \subset \mathbb{R}^s$.*

Given Property 1, it remains to prove that the set of all piecewise constant functions $\mathbb{C}^0(\mathcal{T}, \mathbb{R})$ is dense in $\mathcal{H}(\mathcal{T}, \mathbb{R})$. It is well-known that the set of continuous functions $\mathbb{C}(\mathcal{T}, \mathbb{R})$ is dense in $\mathcal{H}(\mathcal{T}, \mathbb{R})$. Hence, for any $f(\boldsymbol{x}) \in \mathcal{H}(\mathcal{T}, \mathbb{R})$, there exists a continuous function $y \in \mathbb{C}(\mathcal{T}, \mathbb{R})$ such that $\|f(\boldsymbol{x}) - y\|_\mathcal{H} < \epsilon/2$. Since $\mathcal{T}$ is compact and $y$ is continuous, $y$ is uniformly continuous. Hence, there exists an $h > 0$, such that for all $t_1, t_2 \in \mathcal{T}$ with $|t_1 - t_2| < h$, we have $|y(t_1) - y(t_2)| < \epsilon/(2\sqrt{|\mathcal{T}|})$, where $|\mathcal{T}|$ is the measure (length) of $\mathcal{T}$. Let $\{\mathcal{T}_j\}_{j=1}^M$ be a partition of the domain $\mathcal{T}$ with $\max\{|\mathcal{T}_j| : 1 \leq j \leq M\} < h$. Define $f_c(\boldsymbol{x})(t) = \sum_{j=1}^M y(t_j)\delta(t \in \mathcal{T}_j)$, where $t_j$ is an arbitrary point in $\mathcal{T}_j$. Then we have $\|y - f_c(\boldsymbol{x})\|_\mathcal{H} < \epsilon/2$, and it follows that $\|f(\boldsymbol{x}) - f_c(\boldsymbol{x})\|_\mathcal{H} < \epsilon$. Since $\epsilon > 0$ is arbitrary, the set of piecewise constant functions is dense in $\mathcal{H}(\mathcal{T}, \mathbb{R})$.

### B.2 Non-Euclidean Universal Approximation

Let $\mathcal{F} = \{f : \mathbb{R}^s \mapsto \mathcal{H}(\mathcal{T}, \mathcal{M})\}$ denote a family of continuous mappings, where $\mathcal{M}$ is a complete connected Riemannian manifold with metric $d_\mathcal{M}$. We consider the class of feedforward neural networks $\mathcal{NN}_{J,m}$ with $J + 1$ layers and at most $m$ neurons per layer, defined as:

$$\mathcal{NN}_{J,m} = \{T_J \circ a \circ \cdots \circ T_2 \circ a \circ T_1 : \mathbb{R}^s \mapsto \mathcal{H}(\mathcal{T}, \mathbb{R}^p)\},$$

where $T_j$ are affine maps of the form in Eq. (7). Let $\mathcal{NN} = \{\mathcal{NN}_{J,m} : J \geq 2, m \geq 1\}$ denote the class of neural networks of arbitrary depth and width. The notion of convergence on $\mathcal{F}$ is still that of uniform convergence on compact sets of $\mathbb{R}^s$. In analogy with Section B.1, we have the following property:

**Property 2.** *Let $\mathbb{C}^0(\mathcal{T}, \mathcal{M}) \subset \mathcal{H}(\mathcal{T}, \mathcal{M})$ be the set of all piecewise constant functions over the compact interval $\mathcal{T}$. Under certain regularity conditions on the manifold $\mathcal{M}$, for any continuous mapping $f_c : \mathbb{R}^s \mapsto \mathbb{C}^0(\mathcal{T}, \mathcal{M})$, there exist piecewise constant weight parameters $\{\mathcal{W}^{(J)}(t), \mathcal{W}^{(j)}(t), b_j(t)|j = 1, \ldots, J - 1\}$ such that $\sup_{\boldsymbol{x} \in E} \|\rho \circ \mathcal{NN}_{J,m}(\boldsymbol{x}) - f_c(\boldsymbol{x})\|_\mathcal{H} < \epsilon$ for any $\epsilon > 0$. Here, $\rho$ is a readout map that projects the output function from $\mathcal{H}(\mathcal{T}, \mathbb{R}^p)$ onto $\mathcal{H}(\mathcal{T}, \mathcal{M})$; $E \subset \mathbb{R}^s$ is a controlled compact set whose maximal diameter depends on the curvature of $\mathcal{M}$.*

*Proof.* If $\mathcal{M}$ is a Cartan-Hadamard manifold (i.e., a simply connected, complete Riemannian manifold with non-positive sectional curvature), the exponential map $\text{Exp}_{\boldsymbol{y}} : T_{\boldsymbol{y}}\mathcal{M} \mapsto \mathcal{M}$ is a global diffeomorphism, and its inverse (the logarithmic map $\text{Exp}_{\boldsymbol{y}}^{-1} : \mathcal{M} \mapsto \mathbb{R}^p$) is well-defined everywhere. This allows us to linearize manifold-valued functions by mapping them into Euclidean space via $\text{Exp}_{\boldsymbol{y}}^{-1}$ and process them using the neural network class $\mathcal{NN}$. By Corollary 3.14 of [13], since $\text{Exp}_{\boldsymbol{y}}$ is smooth and $\mathcal{NN}$ is dense in the space of continuous mappings from $\mathbb{R}^s$ to $\mathcal{H}(\mathcal{T}, \mathbb{R}^p)$, the class $\{\text{Exp}_{\boldsymbol{y}} \circ \mathcal{NN}_{J,m}|J \geq 2, m \geq 1\}$ is universal in $\mathcal{F}$. That is, for any compact set $E \subset \mathbb{R}^s$ and any continuous mapping $f \in \mathcal{F}$, there exists a neural network $\mathcal{NN}_{J,m}$ such that

$$\sup_{\boldsymbol{x} \in E} d_\mathcal{M}(f(\boldsymbol{x})(t), \text{Exp}_{\boldsymbol{y}} \circ \mathcal{NN}_{J,m}(\boldsymbol{x})(t)) < \epsilon, \tag{8}$$

for any $t \in \mathcal{T}$. The readout map $\text{Exp}_{\boldsymbol{y}}$ acts as a fixed (non-trainable) layer, projecting Euclidean outputs back onto $\mathcal{M}$. The reference point $\boldsymbol{y}$ is typically chosen as the Fréchet mean of the data, minimizing intrinsic variance.

For an arbitrary complete connected Riemannian manifold $\mathcal{M}$, the exponential map $\text{Exp}_{\boldsymbol{y}}$ is only locally diffeomorphic near $\boldsymbol{0} \in T_{\boldsymbol{y}}\mathcal{M}$. If $\mathcal{M}$ is compact, it admits a finite atlas (a finite collection of

smooth coordinate charts covering the manifold). In particular, we can cover it with finitely many logarithmic charts $\{(U_i, \mathrm{Exp}_i^{-1})\}_{i=1}^N$, where each $U_i$ is a geodesic ball of radius $r_i \leq \mathrm{inj}(\mathcal{M})$ (the global injectivity radius),[2] and $\mathrm{Exp}_i^{-1}$ is the locally defined logarithmic map on $U_i$. For any $f \in \mathcal{F}$, define $E_i(f)(t) = \{\boldsymbol{x} \in \mathbb{R}^s : f(\boldsymbol{x})(t) \in U_i\}$ for $i = 1, \cdots, N$. Because each logarithmic map $\mathrm{Exp}_i^{-1}$ is only valid on the open subset $U_i \subseteq \mathcal{M}$, the approximation (8) holds solely for compact subsets of $E_i(f)(t)$, not arbitrary compact subsets of $\mathbb{R}^s$. In particular, there exists a network $\mathcal{NN}_{J,m}$ such that for any compact subset $E \subseteq E_i(f)(t)$,

$$\sup_{\boldsymbol{x} \in E} d_{\mathcal{M}}(f(\boldsymbol{x})(t), \mathrm{Exp}_i \circ \mathcal{NN}_{J,m}(\boldsymbol{x})(t)) < \epsilon.$$

This reflects the intrinsic local nature of exponential charts on general manifolds, contrasting with the global diffeomorphism property in the Cartan-Hadamard case.

$\square$

The proof of Property 2 provides an explicit construction for the readout map $\rho$. Specifically, since $\mathcal{M}$ is compact, there exists a finite set of anchor points $\{\boldsymbol{y}_1, \ldots, \boldsymbol{y}_N\} \subset \mathcal{M}$ such that the geodesic balls $B_{r_i}(\boldsymbol{y}_i)$ cover $\mathcal{M}$, where each radius $r_i$ is upper-bounded by the injectivity radius $\mathrm{inj}_{\mathcal{M}}(\boldsymbol{y}_i)$. On each ball, the logarithmic map $\mathrm{Exp}_{\boldsymbol{y}_j}^{-1}$ defines a smooth coordinate chart, and the transition maps $\mathrm{Exp}_{\boldsymbol{y}_j}^{-1} \circ \mathrm{Exp}_{\boldsymbol{y}_i}$ are smooth diffeomorphisms on overlaps $B_{r_i}(\boldsymbol{y}_i) \cap B_{r_j}(\boldsymbol{y}_j)$. The readout map $\rho$ is then constructed as the collection of exponential maps $\{\mathrm{Exp}_{\boldsymbol{y}_i}\}_{i=1}^N$, which project Euclidean outputs from $\mathcal{H}(\mathcal{T}, \mathbb{R}^p)$ back to $\mathcal{H}(\mathcal{T}, \mathcal{M})$ patchwise.

Our proof of Property 2 generalizes the framework of [13] to arbitrary compact, connected Riemannian manifolds, leveraging local exponential charts to construct a patchwise universal approximator. Unlike the global diffeomorphism property of Cartan-Hadamard manifolds, our approach accounts for the intrinsic locality of general manifolds by partitioning the input space into regions where logarithmic maps are well-defined. Notably, Theorem 6 of [14] underscores the impossibility of global universal approximation for non-Cartan-Hadamard manifolds, aligning with our reliance on local coordinate systems.

## C   Scalar weight regularization

The scalar weights in the two MLPs are regularized through batch normalization and dropout techniques, either together or alternatively. We here briefly explain the dropout and batch normalization techniques; more information can be found in the original publications [28, 11].

The feedforward operation of each hidden layer in Figure 1 can be broken down into two layers: a fully connected layer (with the formula, e.g., $\boldsymbol{x}_1^{(l)} = \boldsymbol{W}\boldsymbol{x}^{(l)} + \boldsymbol{b}$) and then an activation layer (with the formula $\boldsymbol{x}^{(l+1)} = a(\boldsymbol{x}_1^{(l)})$). With batch normalization only, the feedforward operation (over a mini-batch) becomes

$$\boldsymbol{x}_1^{(l)} = \boldsymbol{W}\boldsymbol{x}^{(l)} + \boldsymbol{b},$$
$$\boldsymbol{x}_2^{(l)} = \boldsymbol{\gamma} \circ \mathrm{Norm}(\boldsymbol{x}_1^{(l)}) + \boldsymbol{\eta},$$
$$\boldsymbol{x}^{(l+1)} = a(\boldsymbol{x}_2^{(l)}),$$

where $\circ$ denotes the element-wise product, and $\mathrm{Norm}(\cdot)$ is the normalizing transform (by the batch mean and batch variance). With dropout only, the feedforward operation becomes

$$\boldsymbol{x}_1^{(l)} = \boldsymbol{W}\boldsymbol{x}^{(l)} + \boldsymbol{b},$$
$$\boldsymbol{x}_2^{(l)} = a(\boldsymbol{x}_1^{(l)}),$$
$$\boldsymbol{\psi} \sim \mathrm{Bernoulli}(\tau),$$
$$\boldsymbol{x}^{(l+1)} = \boldsymbol{\psi} \circ \boldsymbol{x}_2^{(l)},$$

where $1 - \tau$ is the dropout rate. If both the dropout and batch normalization techniques are implemented, Figure 4 explains the order of the different layers. Different from the batch normalization

---

[2]Compactness or bounded geometry ensures $\mathrm{inj}(\mathcal{M}) > 0$, allowing finite atlases with geodesic balls of uniform size.

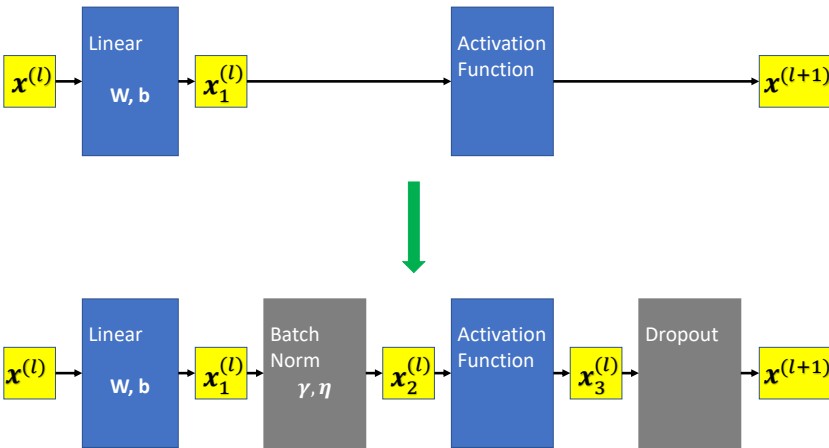

Figure 4: The order of the different layers: the fully connected layer is followed by batch normalization, then non-linear activation, and finally dropout.

technique, where $\boldsymbol{\gamma}$ and $\boldsymbol{\eta}$ are trainable network parameters, the parameter $\tau$ in the dropout technique is a hyper-parameter.

## D   Optimization algorithm: network training

Gradient descent optimizers with adaptive learning rates, e.g., Adam, have become a default method of choice for training feedforward and recurrent neural networks. However, adaptive gradient methods do not generalize well when a regularization term is added to the loss function. Loshchilov and Hutter [18] improved Adam by decaying the weight by a little bit, rather than implementing the gradient of the penalized loss function. However, our objective of regularizing functional weights is to encourage orthogonality, not shrinking their magnitudes. We therefore apply the technique of mini-batch gradient descent with momentum.

Write $\mathcal{L} = \mathcal{L}(\boldsymbol{\theta}; \boldsymbol{y}_{B_i})$, the loss function evaluated on the $i$th batch of the data, where $\boldsymbol{\theta}$ now may include the parameters from bath normalization. The algorithmic framework of the gradient-descent technique is:

> **for** $j = 1, 2, \ldots,$ n_epochs **do**
>     randomly partition the data into mini-batches
>     **for** $i = 1, 2, \ldots,$ n_batches **do**
>         $\boldsymbol{m}_1 = \beta \boldsymbol{m}_0 + (1 - \beta) \nabla_{\boldsymbol{\theta}} \mathcal{L}(\boldsymbol{\theta} = \boldsymbol{\theta}_0; \boldsymbol{y}_{B_i})$
>         $\boldsymbol{\theta}_1 = \boldsymbol{\theta}_0 - \alpha \boldsymbol{m}_1$
>         $\boldsymbol{m}_0 = \boldsymbol{m}_1$
>         $\boldsymbol{\theta}_0 = \boldsymbol{\theta}_1$
>     **end for**
> **end for**

Here, $\alpha > 0$ is the step size, and $0 \leq \beta < 1$ is the momentum weight. The momentum starts at 0. Note that we shuffle the data after every epoch. Within each epoch, after updating $\boldsymbol{m}_0$ and $\boldsymbol{\theta}_0$, we pick the next mini-batch, feed it to the FAE, and calculate the mean gradient of the mini-batch $\nabla_{\boldsymbol{\theta}} \mathcal{L}(\boldsymbol{\theta} = \boldsymbol{\theta}_0; \boldsymbol{y}_{B_i})$.

## E   Optimization algorithm: clustering

To obtain $\lambda_{K-1}$, define the index set $\mathcal{I}_k = \{i : u_i(\lambda_K) = \ddot{u}_k, 1 \leq i \leq n\}$, for $k = 1, \ldots, K$. We then re-formulate the objective function $\mathcal{L}_1$ to be

$$\mathcal{L}_1(\boldsymbol{\zeta}) = \frac{1}{n} \sum\nolimits_{k=1}^{K} \sum\nolimits_{i \in \mathcal{I}_k} (x_i - \zeta_k)^2 + \lambda \sum\nolimits_{1 \leq k < v \leq K} \ddot{\mathbb{s}}_{kv} |\zeta_k - \zeta_v|,$$

where $\boldsymbol{\zeta} = (\zeta_1, \ldots, \zeta_K)^T$ is the vector of $K$ optimization variables, and $\ddot{\mathbb{s}}_{kv} = \sum_{i \in \mathcal{I}_k} \sum_{j \in \mathcal{I}_v} \mathbb{s}(\boldsymbol{y}_i, \boldsymbol{y}_j)$. Let $\boldsymbol{\zeta}(\lambda) = \arg\min_{\boldsymbol{\zeta} \in \mathbb{R}^K} \mathcal{L}_1(\boldsymbol{\zeta})$ denote the solution, again a function of the parameter $\lambda$. Apparently, when $\lambda = \lambda_K$, the vector $\ddot{\boldsymbol{u}} = (\ddot{u}_1, \ldots, \ddot{u}_K)^T$ is the minimizer of the function $\mathcal{L}_1(\boldsymbol{\zeta})$, and hence we have

$$0 = \frac{\partial \mathcal{L}_1}{\partial \zeta_k}(\ddot{\boldsymbol{u}}) = \frac{2}{n}(|\mathcal{I}_k|\ddot{u}_k - \sum_{i \in \mathcal{I}_k} x_i) + \lambda_K \sum_{v=1}^{K} \ddot{\mathbb{s}}_{kv} \times \operatorname{sgn}(\ddot{u}_k - \ddot{u}_v),$$

where sgn(0)=0. In fact, the above equation is valid for any value of $\lambda$ between the current breakpoint $\lambda_K$ and the next breakpoint $\lambda_{K-1}$, and therefore we can write

$$\zeta_k(\lambda) = \frac{1}{|\mathcal{I}_k|}\left[\sum_{i \in \mathcal{I}_k} x_i - \frac{n}{2}\lambda \sum_{v=1}^{K} \ddot{\mathbb{s}}_{kv} \times \operatorname{sgn}(\ddot{u}_k - \ddot{u}_v)\right], \quad \lambda_K \leq \lambda < \lambda_{K-1}. \tag{9}$$

We now can conclude that, when $\lambda_K \leq \lambda < \lambda_{K-1}$ and $i \in \mathcal{I}_k$, we have $u_i(\lambda) = \zeta_k(\lambda)$, and that $\zeta_k(\lambda)$ is a line segment, starting at $\ddot{u}_k$ and with the constant slope

$$\frac{\partial \zeta_k}{\partial \lambda} = -\frac{n}{2|\mathcal{I}_k|}\sum_{v=1}^{K} \ddot{\mathbb{s}}_{kv} \times \operatorname{sgn}(\ddot{u}_k - \ddot{u}_v). \tag{10}$$

Equations (9) and (10) are valid for any value of $\lambda$ as long as the index sets $\{\mathcal{I}_k\}_{k=1}^{K}$ remain unchanged. However, two things could happen at a breakpoint: two index sets get merged into one, or an index set $\mathcal{I}_k$ splits into two. That is, if currently $u_i(\lambda_K) = u_j(\lambda_K)$, then it could happen that $u_i(\lambda) \neq u_j(\lambda)$ for certain $\lambda(> \lambda_K)$. We assume that, at the breakpoints of the solution path, the index sets can never split.[3] To determine the value of the next breakpoint $\lambda_{K-1}$, for any pair of $(\ddot{u}_k, \ddot{u}_v)$, define

$$\triangle_{kv} = (\ddot{u}_k - \ddot{u}_v)\left[\frac{\partial \zeta_v(\lambda)}{\partial \lambda} - \frac{\partial \zeta_k(\lambda)}{\partial \lambda}\right]^{-1}.$$

Then $\triangle_{kv} + \lambda_K$ is the value for $\lambda$ at which the two paths $\zeta_k(\lambda)$ and $\zeta_v(\lambda)$ will be merged together, assuming that no other merge occurs before that. A negative value of $\triangle_{kv}$ indicates that the two paths $\zeta_k(\lambda)$ and $\zeta_v(\lambda)$ are actually moving apart for increasing $\lambda$ and hence will be ignored. Therefore, the next breakpoint is

$$\lambda_{K-1} = \min\{\triangle_{kv}: \triangle_{kv} > 0\} + \lambda_K. \tag{11}$$

If $\triangle_{12} = \min\{\triangle_{kv}: \triangle_{kv} > 0\}$, then the minimizer $\boldsymbol{u}(\lambda_{K-1})$ of $\mathcal{L}_1(\boldsymbol{u})$ for $\lambda = \lambda_{K-1}$ will have $K-1$ unique values, and the two clusters $\{x_i : i \in \mathcal{I}_1\}$ and $\{x_i : i \in \mathcal{I}_2\}$ will be merged into one.

We have explained above how the solution $\boldsymbol{u}(\lambda)$ evolves from the current breakpoint to the next. Now given any (warm) start $\lambda^+ > 0$, let $\boldsymbol{u}(\lambda^+)$ denote the minimizer of $\mathcal{L}_1(\boldsymbol{u})$, obtained by any convex optimization algorithm. (In our code implementation, we adopt the FISTA algorithm developed by Beck and Teboulle [2].) Let the unique values in $\boldsymbol{u}(\lambda^+)$ be denoted by $\{\ddot{u}_1, \ldots, \ddot{u}_K\}$. Exploiting the piecewise linearity of the evolving paths, we can quickly determine the next breakpoint: (1) determine the index sets: $\mathcal{I}_k = \{i : u_i(\lambda^+) = \ddot{u}_k, 1 \leq i \leq n\}$; (2) calculate the affinity weights: $\ddot{\mathbb{s}}_{kv} = \sum_{i \in \mathcal{I}_k} \sum_{j \in \mathcal{I}_v} \mathbb{s}(\boldsymbol{y}_i, \boldsymbol{y}_j)$; (3) calculate the derivatives: $\frac{\partial \zeta_k}{\partial \lambda} = -\frac{n}{2|\mathcal{I}_k|}\sum_{v=1}^{K} \ddot{\mathbb{s}}_{kv} \times \operatorname{sgn}(\ddot{u}_k - \ddot{u}_v)$. Then the next breakpoint is $\lambda_{K-1} = \min\{\triangle_{kv}: \triangle_{kv} > 0\} + \lambda^+$. Apparently, at trivial additional computations, our path-following homotopy algorithm will produce a hierarchy, with the $K$ clusters being repeatedly merged.

## F  Supplementary materials for the experiments

### F.1  Details on algorithm configuration

The same basis family and evaluation grid are used for smoothing across all algorithms to maintain consistency. For all baseline methods, we run each method with the number of clusters ranging from 2 to 10 (extended to 20 for the Fungi dataset), ensuring that the true number of clusters is included in this range. We then report the best clustering performance based on ARI and AMI. In contrast, FAEclust determines the number of clusters in a fully data-driven manner using the silhouette score within each forward phase of the joint training and clustering framework.

---

[3]If this assumption is valid, then our path-following homotopy algorithm will produce the exact solution; otherwise, we will interpret our solution as an approximation. Splitting events are infrequent.

1. funHDDC (from R package `funHDDC`, [3]): All six supported models were evaluated: "AkjBkQkDk", "AkjBQkDk", "AkBkQkDk", "ABkQkDk", "AkBQkDk", and "ABQkDk". Additional parameters included initialization via `init = "means"`, a convergence threshold of `threshold = 0.1`, model selection based on `criterion = "bic"`, and a maximum of `itermax = 100` iterations. For each dataset, the final clustering result is given by the model that achieved the lowest BIC value.

2. funclust (from R package `funclust`, [12]): The relevant arguments are set to `thd = 0.05`, `increaseDimension = FALSE`, `hard = FALSE`, `fixedDimension = integer(0)`, `nbInit = 20`.

3. FADPclust (from R package `FADPclust`, [34]): The package consists of two clustering methods `method = "FADP1"` and `method = "FADP2"`. We evaluated both methods using the default parameters `proportion = NULL` and `f.cut = 0.15`.

4. FNN ([8]): The FNN architecture is $\boldsymbol{y} \xrightarrow{\boldsymbol{U}^{(1)}(s,\tau)\in\mathcal{H}(\mathcal{T}\times\mathcal{T},\mathbb{R})} \boldsymbol{h}^{(1)} \xrightarrow{\boldsymbol{U}^{(2)}(s,\tau)\in\mathcal{H}(\mathcal{T}\times\mathcal{T},\mathbb{R})} \boldsymbol{h}^{(2)} \xrightarrow{\boldsymbol{V}^{(1)}(t)\in\mathcal{H}(\mathcal{T},\mathbb{R})} \boldsymbol{x}^{(1)} \rightarrow \boldsymbol{x} \rightarrow \hat{\boldsymbol{x}}^{(1)} \xrightarrow{\boldsymbol{W}^{(1)}(t)\in\mathcal{H}(\mathcal{T},\mathbb{R})} \hat{\boldsymbol{y}}$. $\boldsymbol{U}^{(1)}(s,\tau)$ and $\boldsymbol{U}^{(2)}(s,\tau)$ are Legendre-basis convolution kernels. The first hidden layer is $\boldsymbol{h}^{(1)}(s) = a\left( \int_{\mathcal{T}} \boldsymbol{U}^{(1)}(s,\tau)\boldsymbol{y}(\tau)d\tau + \boldsymbol{b}^{(1)}(s) \right)$, where $a$ is ELU activation. Channel sizes are 32 for $\boldsymbol{U}^{(1)}$ and 16 for $\boldsymbol{U}^{(2)}$. $\boldsymbol{V}^{(1)}(t)$ is a Fourier–basis functional producing a 16-dimensional vector $\boldsymbol{x}^{(1)}$. The block $\boldsymbol{x}^{(1)} \rightarrow \boldsymbol{x} \rightarrow \hat{\boldsymbol{x}}^{(1)}$ is a 3-layer MLP, each layer with 16 neurons. The output layer is linear without bias: $\hat{\boldsymbol{y}}(t) = \boldsymbol{W}^{(1)}(t)\hat{\boldsymbol{x}}^{(1)}$. Learned embeddings are clustered by $k$-means and evaluated via AMI/ARI.

   Settings: `latent_dim = 16`, `conv_channels = c(32, 16)`, `conv_basis = "Legendre"`, `dense_basis = "Fourier"`, `activation = "ELU"`, `padding = "same"`, `optimizer = "Adam"`, `lr = 1e-3`, `loss = "MSE"`, `epochs = 200`, `kmeans_n_init = 20`.

5. FAE ([10]): The FAE architecture is $\boldsymbol{y} \xrightarrow{\boldsymbol{W}^{(1)}(t)\in\mathcal{H}(\mathcal{T},\mathbb{R})} \boldsymbol{x}^{(1)} \rightarrow \boldsymbol{x} \rightarrow \hat{\boldsymbol{x}}^{(1)} \xrightarrow{\boldsymbol{W}^{(1)}(t)\in\mathcal{H}(\mathcal{T},\mathbb{R})} \hat{\boldsymbol{y}}$. The functional layer uses tanh activation, and MLP layers use ELU activation; the output layer is linear, no bias. Functional weights are represented via basis expansions (B-splines). Learned embeddings are clustered by $k$-means and evaluated via AMI/ARI.

   Settings: `basis = "B-spline"`, `basis_dim = 10`, `latent_dim = 16`, `layers = 32, 16`, `activation = "tanh"`, `optimizer = "ELU"`, `lr = 1e-3`, `epochs = 200`, `loss = "MSE"`, `kmeans_n_init = 20`.

6. VANO (`VANO`, [25]): VANO is trained with Adam on reconstruction MSE plus a $\beta$-weighted KL term. Layer widths are $\{64, 64, 16, 64, 64\}$. Encoder means are clustered by $k$-means and evaluated with AMI/ARI.

   Settings: `latent_dim = 16`, `hidden_sizes = c(64, 64)`, `activation = "GELU"`, `reparameterization = TRUE`, `optimizer = "Adam"`, `lr = 1e-3`, `beta = 1e-4`, `epochs = 200`, `kmeans_n_init = 20`.

7. FAEclust (`FAEclust`: FAEclust shares hyperparameters with standard autoencoders, including the number of training epochs (`epochs`), learning rate (`lr`), batch size (`batch_size`), and dropout rate (`tau`). The network architecture is defined by the list `layers`, where the first entry and last three entries specify functional layers, and the middle entries form an MLP autoencoder. Functional weights and biases are parameterized using a basis family `network_basis` of size `l`. The network's objective function includes two regularization terms: an orthogonality penalty on the encoder's functional weights (`lambda_e`) and a sparsity penalty on the decoder's functional weights and biases (`lambda_d`). All hyperparameters – `epochs`, `lr`, `batch_size`, `tau`, `layers`, `lambda_e`, and `lambda_d` – are optimized via Bayesian optimization over a predefined search space. The objective function in the Bayesian optimization problem is still the integrated objective function $\mathcal{L} = \mathcal{L}_r + \lambda_{\mathrm{w}}\mathcal{L}_{\mathrm{w}} + \lambda_c\mathcal{L}_c$. However, in the context of Bayesian optimization, the variables being optimized are the hyperparameters, not the model weights. In our implementation, we utilized the Optuna package to perform the hyperparameter tuning task.

   In all experiments, both real and simulated, we fixed the `FAEclust` architecture to a seven-layer structure: one functional layer in the encoder, three functional layers in the decoder,

and a three-layer MLP in between. The third layer in `FAEclust` serves as the bottleneck, producing the latent embedding. The maximum number of nodes allowed per layer was constrained to $\{64, 32, 16, 32, 64, 64, 64\}$.

## F.2 Details on the simulated manifold-valued functional datasets

Table 5 summarizes the number of samples, dimensions, time steps, and ground-truth clusters for each simulated functional dataset.

Table 5: Dataset parameters.

| Dataset | # Samples | # Dimensions | Time steps | # Clusters |
|---|---|---|---|---|
| Hypersphere | 100 | 3 | 100 | 2 |
| Hyperbolic | 200 | 2 | 50 | 2 |
| Swiss roll | 300 | 2 | 200 | 4 |
| Lorenz | 100 | 3 | 100 | 3 |
| Pendulum | 200 | 2 | 100 | 4 |

**Hypersphere [19]**   Figure 5 visualizes the clusters on a hypersphere $S^2$ consisting of all unit vectors in $R^3$. We generate trajectories along great circles on the sphere, using different directions for the two clusters. The two clusters differ in period and are phase-shifted randomly, making this a challenging dataset for clustering algorithms that do not perform curve registration.

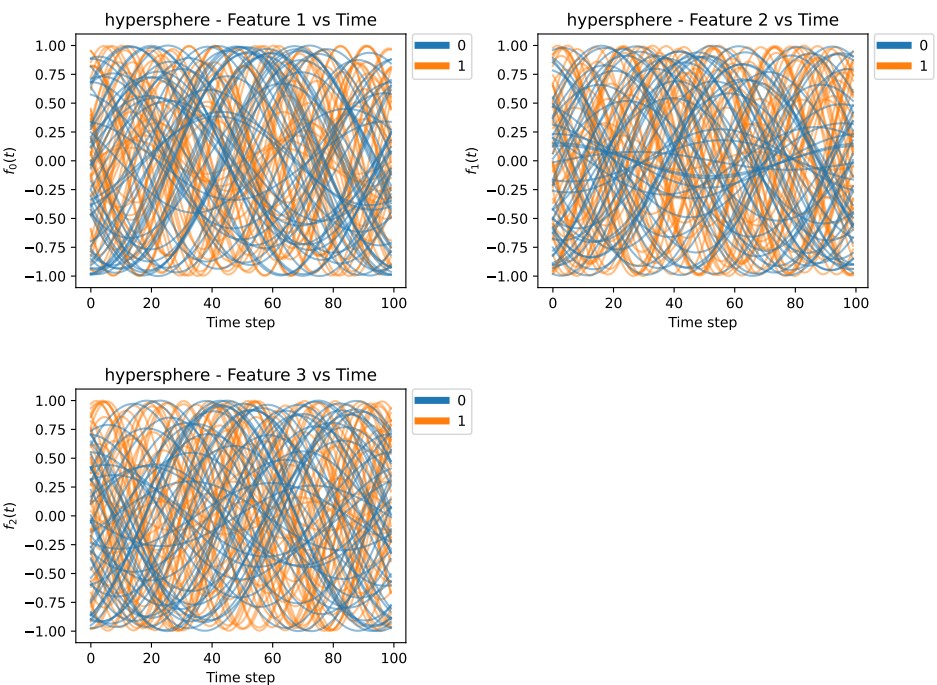

Figure 5: Hypersphere functional data. Trajectories on the surface of a hypersphere $S^2$, with clusters defined by distinct great-circle paths and phase shifts.

**Hyperbolic [20]**   Figure 6 illustrates trajectories in hyperbolic space using the Poincaré ball model, which represents a space of constant negative curvature. We simulate geodesic motion within this ball. One class stays near the center, and the other ventures closer to the boundary.

**Lorenz [17]**   Figure 7 shows time series generated from the Lorenz system, a well-known chaotic system in three dimensions. For certain parameter values (sigma=10, rho=28, beta=8/3), the solutions

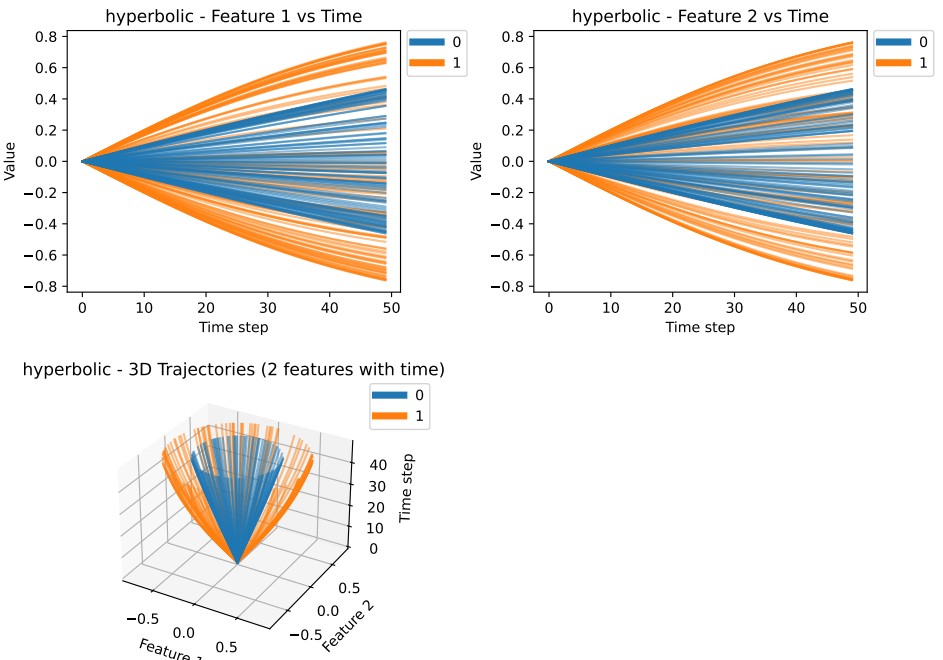

Figure 6: Hyperbolic functional data. Trajectories simulated in the Poincaré ball model of hyperbolic space, with clusters differentiated by proximity to the center or boundary.

approach the butterfly-shaped Lorenz attractor. We simulate three classes: a non-chaotic class with rho=14, and two chaotic classes with rho=21 and rho=28.

**Swiss roll [31]**  Figure 8 depicts the Swiss roll, a classic example of a 2D manifold embedded in 3D. We generate trajectories along the rolled surface, with different clusters corresponding to different vertical ranges.

**Pendulum [30]**  Figure 9 shows pendulum trajectories in cylindrical phase space $(\mathbb{R} \times S^1)$, defined by angular position and angular velocity. We simulate both oscillatory motion (low energy) and full rotations (high energy), resulting in four distinct clusters.

### F.3  ARI tables for the Euclidean and manifold-valued functional datasets

Tables 6 and 7 present the ARI scores for our benchmarks. Table 6 covers the 17 Euclidean functional datasets (see Table 1 for the corresponding AMI results), while Table 7 summarizes the ARI scores for the five manifold-based scenarios (cf. Table 2).

For the manifold-valued FD simulation, Figure 10 shows boxplots of the number of clusters identified over 100 repetitions for each manifold type. The boxplots indicate that `FAEclust` consistently finds the true number of clusters with low run-to-run variability across 100 repetitions.

### F.4  Experiments on time-warped functional data

Figures 11 and 12 respectively illustrate the original and time-warped functional dataset in one repetition, with different clusters shown in different colors.

Tables 8 and 9 summarize the AMI and ARI performance, respectively, of the seven baseline methods on the time-warped datasets. Overall, `FAEclust` delivers strong, robust clustering on time-warped functional data, outperforming baselines in most scenarios. These results highlight the advantages of using a shape-informed metric for clustering problems where phase variation is a nuisance.

For the 12 simulation scenarios, Figure 13 shows boxplots of the number of clusters identified over 100 repetitions for each simulation scenario. The dashed line in each panel indicates the true numbers

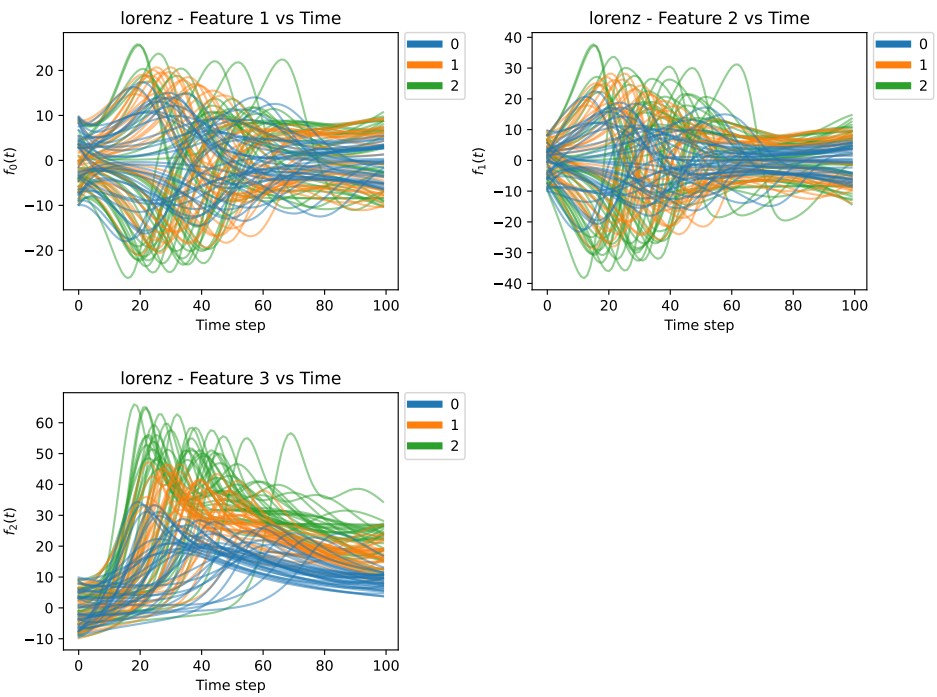

Figure 7: Lorenz functional data. Trajectories generated from the Lorenz system, illustrating chaotic dynamics across different parameter settings.

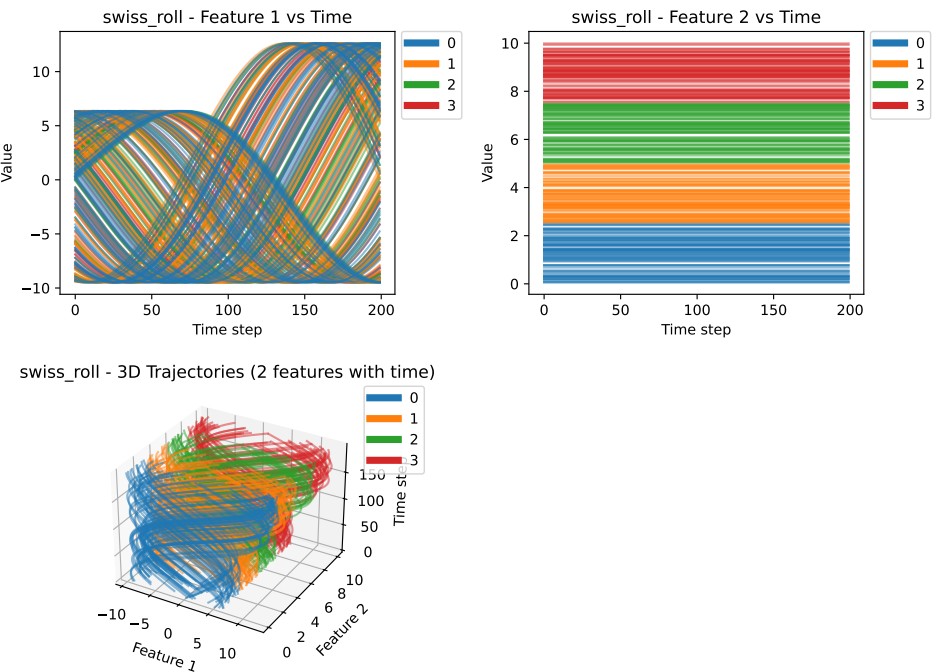

Figure 8: Swiss roll functional data. Trajectories on a 2D manifold embedded in 3D space, with different colors indicating different clusters.

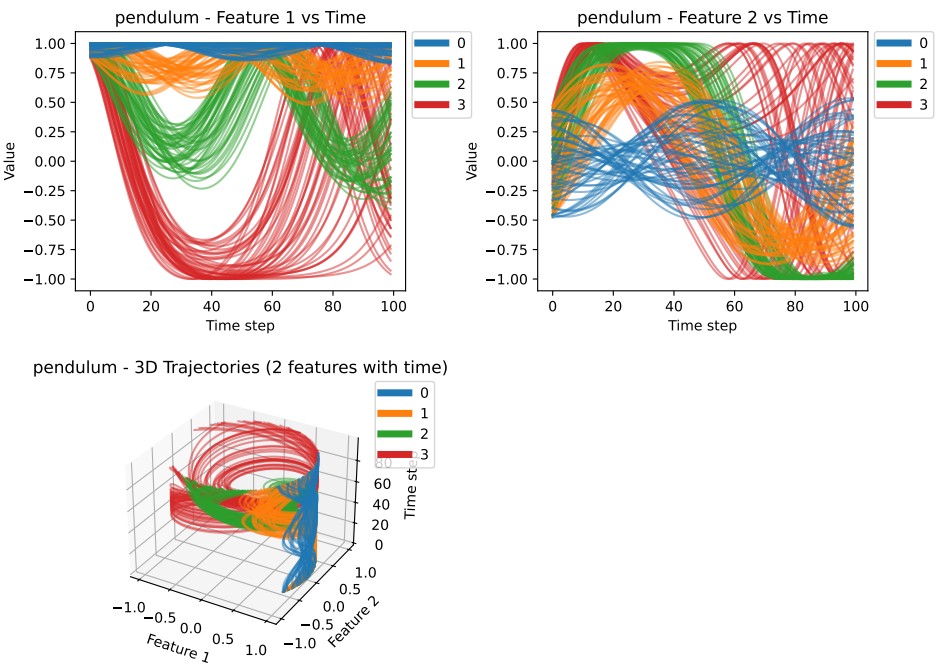

Figure 9: Pendulum functional data. Trajectories in cylindrical phase space $(\mathbb{R} \times S^1)$, with colors indicating different clusters.

Table 6: ARI scores for the 17 Euclidean functional datasets.

| Dataset | funHDDC | funclust | FADP1 | FADP2 | FNN | FAE | VANO | FAEclust |
|---|---|---|---|---|---|---|---|---|
| BirdChicken | 0.068 | 0.101 | 0.026 | 0.068 | **0.232** | 0.169 | 0.148 | 0.227 |
| CBF | 0.343 | 0.030 | 0.438 | 0.234 | 0.329 | 0.339 | 0.643 | **0.709** |
| Chinatown | 0.198 | 0.198 | 0.080 | 0.124 | 0.174 | 0.093 | 0.256 | **0.282** |
| DSR | 0.750 | 0.025 | 0.708 | 0.822 | 0.563 | 0.462 | 0.644 | **0.862** |
| ECG200 | **0.186** | 0.158 | 0.129 | 0.144 | 0.134 | 0.137 | 0.119 | 0.164 |
| Fungi | 0.593 | 0.091 | 0.258 | 0.317 | 0.126 | 0.495 | 0.702 | **0.853** |
| Plane | 0.764 | 0.006 | 0.586 | 0.662 | 0.783 | 0.740 | 0.728 | **0.825** |
| Rock | **0.321** | 0.267 | 0.104 | 0.291 | 0.137 | 0.101 | 0.263 | 0.297 |
| Symbols | 0.664 | 0.000 | 0.362 | 0.576 | 0.630 | 0.638 | 0.676 | **0.682** |
| Blink | 0.369 | 0.039 | 0.164 | 0.153 | 0.418 | 0.449 | 0.442 | **0.563** |
| BM | 0.306 | 0.017 | 0.146 | 0.344 | 0.334 | **0.626** | 0.528 | 0.497 |
| EOS | 0.160 | 0.004 | 0.067 | 0.072 | 0.056 | 0.175 | 0.131 | **0.193** |
| Epilepsy | 0.095 | 0.014 | 0.049 | 0.166 | 0.180 | 0.137 | 0.193 | **0.328** |
| ERing | 0.641 | 0.031 | 0.172 | 0.651 | 0.576 | **0.677** | 0.669 | 0.591 |
| FM | 0.001 | 0.004 | 0.003 | 0.002 | 0.105 | **0.143** | 0.077 | 0.139 |
| JV | 0.810 | 0.054 | 0.182 | 0.344 | 0.085 | 0.843 | **0.887** | 0.879 |
| SWJ | 0.191 | 0.029 | 0.183 | 0.048 | 0.135 | 0.122 | **0.288** | 0.261 |

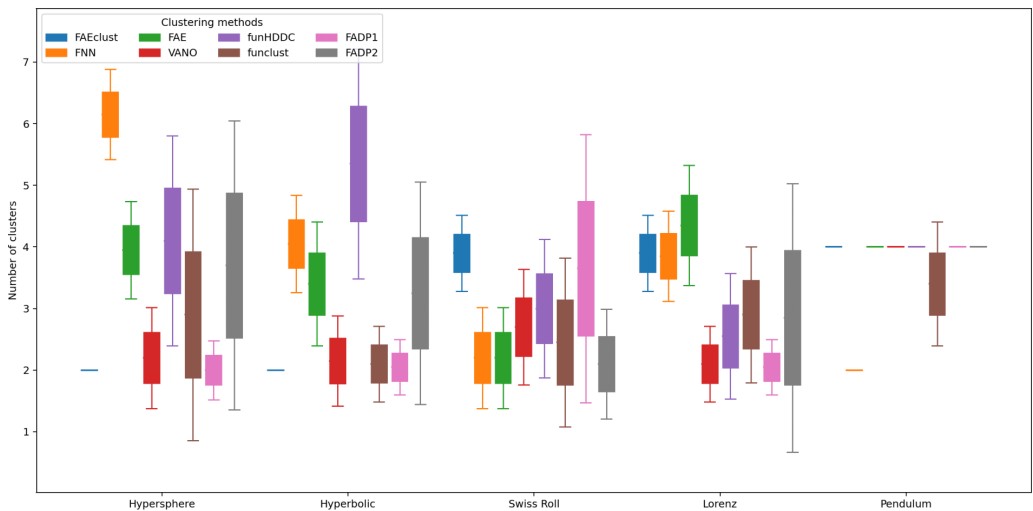

Figure 10: Per manifold scenario, 100 FD datasets were generated and analyzed using the eight clustering methods; the boxplots (one per method) summarize the number of clusters identified across the 100 runs. Only `FAEclust` and VANO yield consistent results, with `FAEclust` showing lower run-to-run variability.

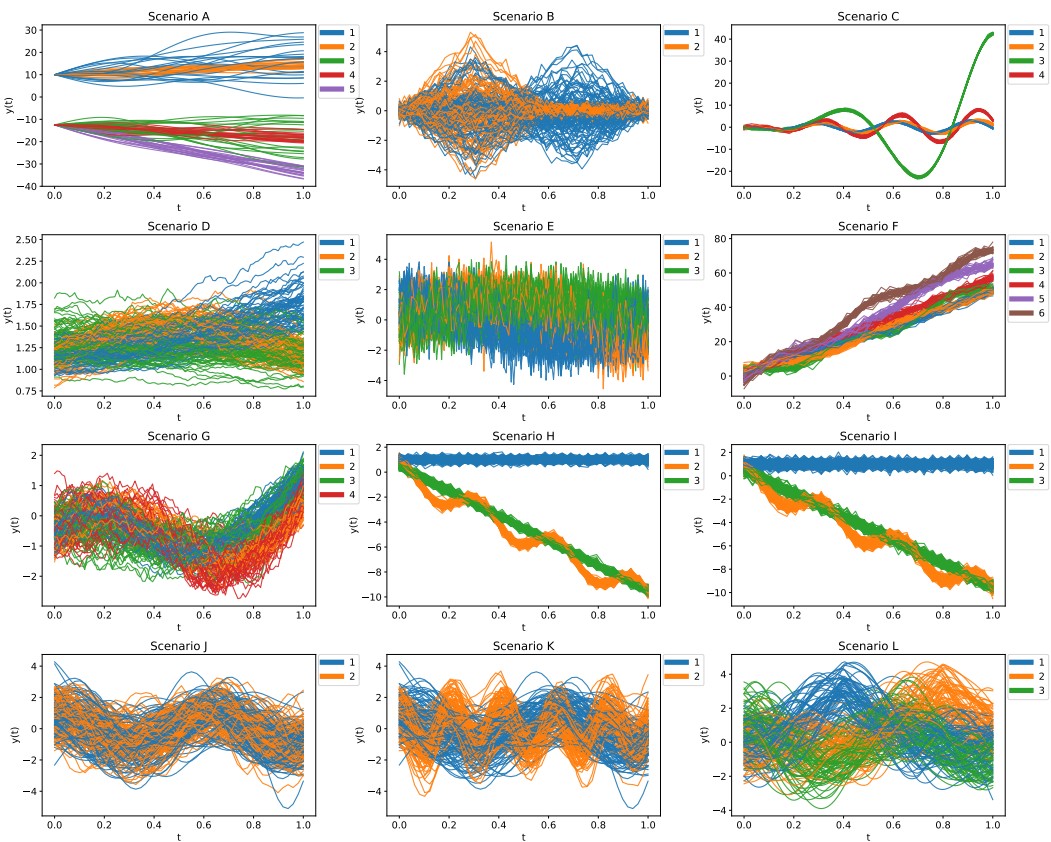

Figure 11: Example of original unwarped functional data from one repetition. Different colors represent different clusters.

Table 7: ARI scores for the five manifold-valued functional datasets. The table reports the mean (top row) and standard deviation (bottom row) of the scores over 100 repetitions.

| Dataset | funHDDC | funclust | FADP1 | FADP2 | FNN | FAE | VANO | FAEclust |
|---|---|---|---|---|---|---|---|---|
| Hypersphere | 0.014 | 0.461 | 0.107 | 0.038 | 0.067 | 0.228 | 0.411 | **0.696** |
| | 0.032 | 0.039 | 0.124 | 0.046 | 0.058 | 0.041 | 0.060 | **0.032** |
| Hyperbolic | 0.005 | 0.007 | 0.001 | 0.001 | 0.004 | 0.058 | 0.213 | **0.744** |
| | 0.014 | 0.030 | 0.008 | 0.008 | **0.003** | 0.010 | 0.035 | 0.037 |
| Swiss roll | 0.069 | 0.097 | **0.271** | 0.109 | 0.016 | 0.057 | 0.142 | 0.205 |
| | 0.034 | 0.032 | 0.054 | 0.071 | 0.026 | **0.018** | 0.045 | 0.049 |
| Lorenz | 0.103 | 0.348 | 0.063 | 0.126 | 0.036 | 0.180 | 0.189 | **0.416** |
| | 0.062 | 0.048 | 0.042 | 0.079 | 0.039 | 0.025 | **0.018** | 0.031 |
| Pendulum | 0.847 | 0.316 | 0.735 | 0.744 | 0.195 | 0.680 | 0.874 | **0.985** |
| | 0.031 | 0.053 | 0.076 | 0.068 | 0.033 | 0.050 | 0.047 | **0.009** |

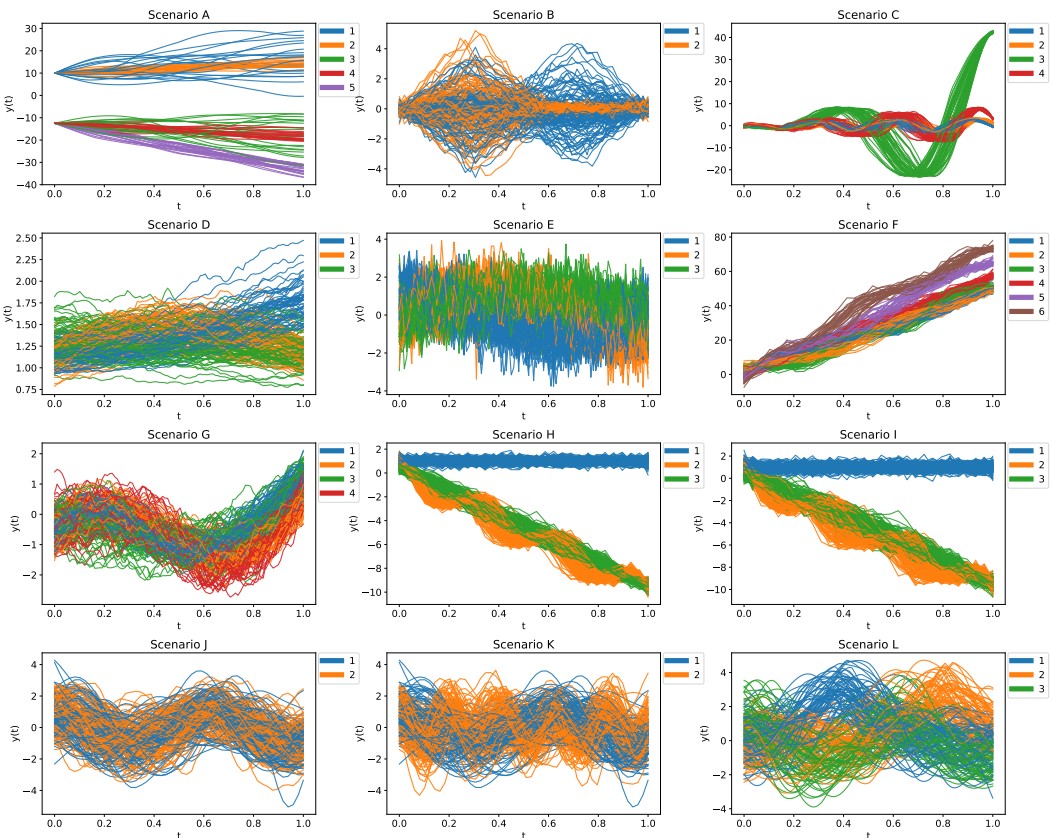

Figure 12: Example of time-warped functional data from one repetition. Different colors represent different clusters.

Table 8: AMI scores of seven baseline methods on time-warped datasets. For each simulation scenario, the table shows the mean (top row) and standard deviation (bottom row) of AMI scores across 100 independent runs.

| Scenario | funHDDC | funclust | FADP1 | FADP2 | FNN | FAE | VANO |
|---|---|---|---|---|---|---|---|
| A | 0.506 | 0.407 | 0.411 | 0.402 | 0.470 | 0.436 | 0.654 |
|   | 0.101 | 0.011 | 0.012 | 0.026 | 0.064 | 0.019 | 0.029 |
| B | 0.096 | 0.013 | 0.009 | 0.097 | 0.001 | 0.000 | 0.005 |
|   | 0.029 | 0.024 | 0.025 | 0.069 | 0.006 | 0.005 | 0.017 |
| C | 0.542 | 0.199 | 0.303 | 0.402 | 0.500 | 0.457 | 0.571 |
|   | 0.078 | 0.103 | 0.094 | 0.013 | 0.006 | 0.025 | 0.029 |
| D | 0.299 | 0.069 | 0.290 | 0.321 | 0.058 | 0.057 | 0.489 |
|   | 0.051 | 0.093 | 0.073 | 0.071 | 0.029 | 0.025 | 0.080 |
| E | 0.469 | 0.687 | 0.578 | 0.878 | 0.868 | 0.890 | 0.871 |
|   | 0.094 | 0.084 | 0.010 | 0.024 | 0.029 | 0.015 | 0.030 |
| F | 0.664 | 0.277 | 0.338 | 0.324 | 0.308 | 0.459 | 0.664 |
|   | 0.049 | 0.078 | 0.061 | 0.066 | 0.051 | 0.044 | 0.045 |
| G | 0.285 | 0.005 | 0.202 | 0.149 | 0.280 | 0.253 | 0.206 |
|   | 0.036 | 0.011 | 0.045 | 0.061 | 0.043 | 0.044 | 0.062 |
| H | 0.592 | 0.579 | 0.366 | 0.579 | 0.641 | 0.724 | 0.861 |
|   | 0.047 | 0.013 | 0.117 | 0.032 | 0.032 | 0.026 | 0.058 |
| I | 0.592 | 0.579 | 0.336 | 0.560 | 0.635 | 0.658 | 0.672 |
|   | 0.052 | 0.022 | 0.109 | 0.019 | 0.030 | 0.015 | 0.057 |
| J | 0.001 | 0.001 | 0.000 | 0.000 | 0.000 | 0.000 | 0.001 |
|   | 0.003 | 0.008 | 0.005 | 0.005 | 0.004 | 0.005 | 0.006 |
| K | 0.366 | 0.003 | 0.200 | 0.049 | 0.129 | 0.097 | 0.057 |
|   | 0.085 | 0.004 | 0.097 | 0.061 | 0.104 | 0.101 | 0.065 |
| L | 0.381 | 0.001 | 0.514 | 0.494 | 0.728 | 0.714 | 0.648 |
|   | 0.047 | 0.006 | 0.076 | 0.060 | 0.064 | 0.061 | 0.129 |

of clusters. Again, `FAEclust` performs best: it consistently recovers the true cluster number and exhibits less variability in the identified counts than competitors. Even in Scenario J, where variability increases, it remains the only method to find the true cluster count.

## F.5 Limitations

In all the experiments above, dynamic time warping computations were performed on two 2.90 GHz CPUs (each with 16 cores), and the resulting distance matrices were used to train the FAE network on an NVIDIA Quadro RTX 5000 GPU with 16 GB of VRAM.

The main limitations of our model are its sensitivity to hyperparameter settings and its higher computational cost. Even with fixed hyperparameter values, `FAEclust` takes significantly longer to run than most existing FD clustering methods. For example, on the "Plane" dataset from the UCR repository – comprising 210 samples, each of length 144, and 7 clusters – funHDDC completes in 24 seconds, FADP1 in 18 seconds, FADP2 in 23 seconds, funclust in 284 seconds, FNN in 48 s, FAE in 31 s, VANO in 51 s, while `FAEclust` requires 220 seconds. This makes it roughly ten times slower than the fastest baselines (funHDDC, FADP1, FADP2, and FAE). In practical applications, users can first apply a fast baseline method such as FADP2 to obtain initial clustering results. These results can then serve as a warm start for our joint network training and clustering framework (Figure 2).

To enhance clustering performance, we optionally include a Bayesian optimization step for hyperparameter tuning. While more sample-efficient than grid or random search, this step further increases the overall runtime. In practical applications, users must weigh FAEclust's improved modeling flexibility and accuracy – achieved through its deep learning foundation and shape-aware objective – against this added computational burden.

Table 9: ARI scores of seven baseline methods on randomly generated time-warped datasets. For each simulation scenario, the table shows the mean (top row) and standard deviation (bottom row) of ARI scores across 100 independent runs.

| Scenario | funHDDC | funclust | FADP1 | FADP2 | FNN | FAE | VANO |
|---|---|---|---|---|---|---|---|
| A | 0.423 | 0.362 | 0.365 | 0.372 | 0.334 | 0.335 | 0.524 |
|   | 0.108 | 0.010 | 0.024 | 0.030 | 0.074 | 0.018 | 0.027 |
| B | 0.063 | 0.016 | 0.009 | 0.074 | 0.001 | 0.001 | 0.005 |
|   | 0.029 | 0.032 | 0.019 | 0.059 | 0.008 | 0.006 | 0.013 |
| C | 0.449 | 0.151 | 0.281 | 0.330 | 0.436 | 0.344 | 0.502 |
|   | 0.086 | 0.095 | 0.088 | 0.018 | 0.028 | 0.028 | 0.033 |
| D | 0.284 | 0.080 | 0.267 | 0.291 | 0.053 | 0.050 | 0.456 |
|   | 0.063 | 0.101 | 0.086 | 0.076 | 0.030 | 0.023 | 0.087 |
| E | 0.438 | 0.660 | 0.571 | 0.869 | 0.828 | 0.864 | 0.850 |
|   | 0.122 | 0.096 | 0.016 | 0.031 | 0.036 | 0.029 | 0.017 |
| F | 0.610 | 0.208 | 0.282 | 0.243 | 0.188 | 0.339 | 0.580 |
|   | 0.048 | 0.070 | 0.057 | 0.052 | 0.046 | 0.051 | 0.043 |
| G | 0.294 | 0.003 | 0.210 | 0.155 | 0.228 | 0.203 | 0.161 |
|   | 0.059 | 0.008 | 0.063 | 0.074 | 0.036 | 0.033 | 0.053 |
| H | 0.524 | 0.551 | 0.302 | 0.551 | 0.554 | 0.701 | 0.826 |
|   | 0.053 | 0.019 | 0.101 | 0.026 | 0.023 | 0.060 | 0.059 |
| I | 0.497 | 0.571 | 0.265 | 0.547 | 0.551 | 0.554 | 0.631 |
|   | 0.081 | 0.025 | 0.120 | 0.020 | 0.017 | 0.048 | 0.080 |
| J | 0.001 | 0.001 | 0.003 | 0.005 | 0.001 | 0.001 | 0.002 |
|   | 0.004 | 0.009 | 0.005 | 0.004 | 0.006 | 0.006 | 0.005 |
| K | 0.346 | 0.001 | 0.195 | 0.047 | 0.165 | 0.127 | 0.061 |
|   | 0.134 | 0.005 | 0.119 | 0.070 | 0.130 | 0.129 | 0.065 |
| L | 0.439 | 0.001 | 0.560 | 0.532 | 0.667 | 0.652 | 0.622 |
|   | 0.040 | 0.007 | 0.069 | 0.059 | 0.067 | 0.062 | 0.079 |

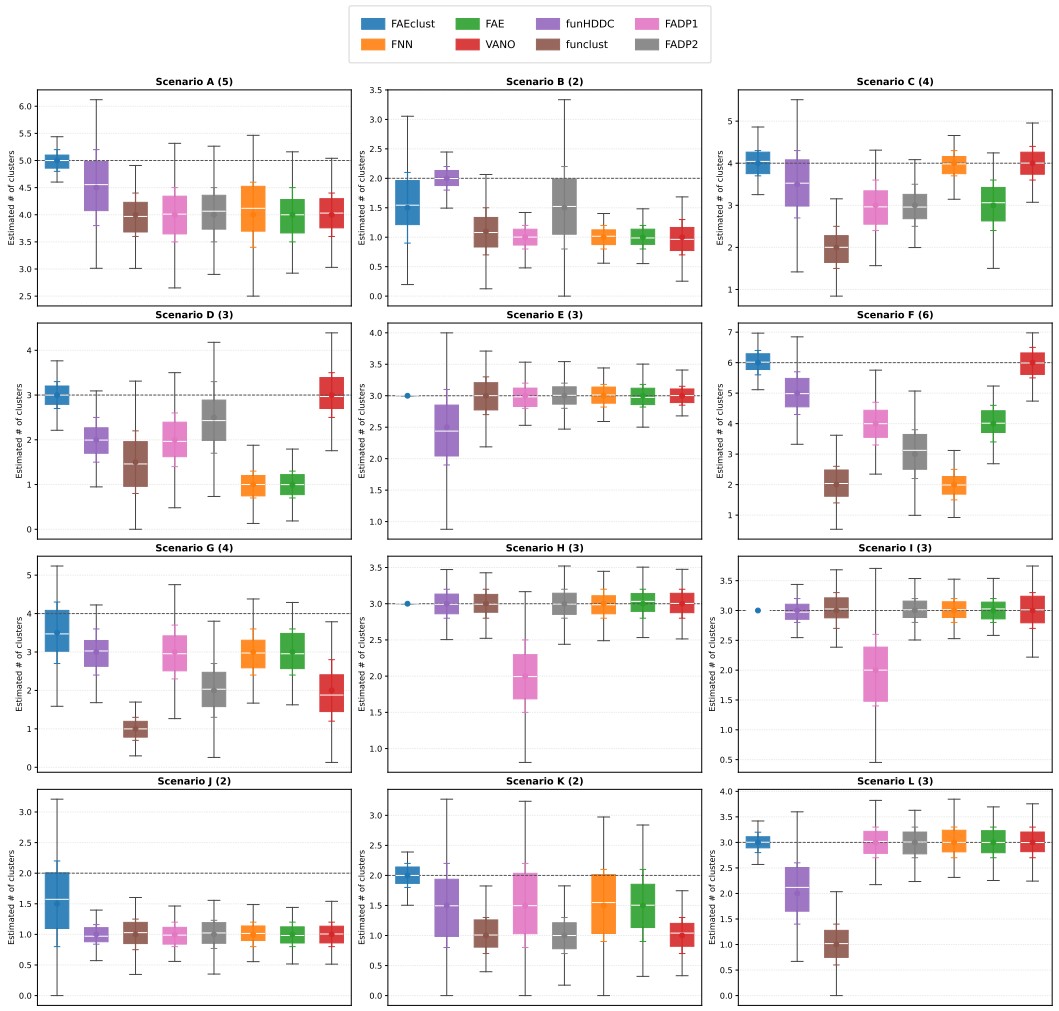

Figure 13: Per simulation scenario, 100 FD datasets were generated and analyzed using the eight clustering methods; the boxplots (one per method) summarize the number of clusters identified across the 100 runs. The numbers in the parentheses are the true numbers of clusters.

