# OpenReview forum: "Shape-Informed Clustering of Multi-Dimensional Functional Data via Deep Functional Autoencoders"
_NeurIPS.cc/2025/Conference — NeurIPS 2025 poster_

### Official Review · Reviewer_5zSQ · 2025-06-15

**Clarity:** 3
**Significance:** 2
**Originality:** 3
**Rating:** 5
**Confidence:** 3

**Summary:**

This work develops a clustering method for multivariate functional data based on deep functional autoencoder. It incorporates layers with functional weights in the network and clustering loss in the objective function to adapt classic autoencoder for the functional clustering task. Additionally, a shape-informed SRV metric is used in the clustering objective function to account for phase variation in functional data. A universal approximation theorem is presented for reconstruction accuracy and numerical experiments confirm the effectiveness of the proposed method.

**Questions:**

1. Could you explain briefly the methods used as the benchmarks? For example, which category of the two types of clustering methods (line 21-24) do they fall into? Are they deterministic or probabilistic?
2. Could you explain how do you explore the number of clusters (line 535-536)? Does it recover the true number of clusters (if available)? Do any of the benchmarks include choosing the number of clusters in their framework?
3. Could you further clarify your theoretical result based on the questions in Limitations, first point (see below)?
4. Could you provide additional comparisons based on the questions in Limitations, second point (see below)?

**Ethical Concerns:**

["NO or VERY MINOR ethics concerns only"]

**Final Justification:**

The rebuttal answers my questions. The paper is well-written with sufficient amount of novelty, supported by comprehensive theoretical and empirical results.

**Limitations:**

1. It may be better to provide readers some intuitions and clarifications before and after Theorem 1. For example, what question it tries to answer?  Why it is useful? How does it connect and compare to classic universal approximation theorem? Also, the readout map $\rho$ is not clearly defined in the theorem.
2. $\textbf{Shape-informed clustering}$ is the first phrase in the title, but only the clustering objective function in Section 4.1 is relevant to it. Is this one of the designs in your method that you would like to highlight? If it is the case, maybe it would be better to put more efforts in motivating and demonstrating the merits of introducing a shape-informed metric, both throughout writing and in experiments. For example, if a classic $\mathbb{L}^2$ metric is used, how would the results change in Table 1? If phase variation is introduced (Appendix E.3), how would the benchmark methods perform?
3. In general it would be better to report uncertainty for evaluation metrics (Table 1 and 2), e.g., repetition over data splits.

**Paper Formatting Concerns:**

None.

**Quality:**

3

**Strengths And Weaknesses:**

Strength:

1. The main paper has very clear structure and notation and is very well-written. The appendix is very comprehensive and include sufficient amount of details that are not included in the main paper.
2. The idea of using autoencoder for functional clustering is very interesting and novel. The method is flexible and well-designed for this task.

Weaknesses:

1. The connection between theoretical result (existence of a good functional network) and practical design of the method (how to find it) is relatively weak.
2. There are a lot of modeling choices, e.g., penalty tuning, number of basis for functional weights, number of clusters, which can potentially affect the robustness of the method.
3. As pointed out by the authors, the method is computationally expensive.

---

> ### Author Rebuttal · Authors · 2025-07-24
>
> Thank you for these constructive comments, which, together with those from other reviewers, have helped us strengthen the paper’s contribution and clarify key aspects.
>
> **Response to Weakness:** In our response to Reviewer CGH6 (Q1), we presented additional experiments demonstrating the importance of using three functional layers in the decoder (rather than a single layer). These results empirically validate the necessity of multiple functional layers and support Theorem 1. Across all experiments, we fixed the network architecture to 7 layers, which we found sufficient in practice. We then apply Bayesian optimization to select the width of each layer, with the number of nodes per layer constrained to the set {32, 16, 8, 16, 32, 32, 32} as the maximum allowed configuration. In the revised manuscript, we will provide layer-wise information of the trained FAEclust for each dataset.  For more challenging datasets (e.g., manifold-valued data), users may increase the layer widths to further enhance FAEclust’s representational power. Please see our response to Reviewer WLrd for further details and supporting experiments.
>
> Regarding modeling choices, the two penalty parameters $\lambda_w$ and $\lambda_c$,  along with the number of basis functions for the four functional layers, are additional hyperparameters introduced beyond those in traditional autoencoders.  The parameter $\lambda$ in the clustering objective function and the number of clusters are not hyperparameters (please see our first response to Reviewer CGH6 for clarification). To assess robustness, we ran FAEclust 20 times per dataset in all experiments reported above, presenting the mean and standard deviation of ARI and AMI scores. The results (summarized in the tables above) demonstrate that FAEclust is stable and robust across datasets.
>
> **Response to Q1:** Thank you for the comment. In our revision, we will also cite the publications corresponding to each R package in the introduction.
>
> - funHDDC [1]: This method projects functional data onto a large set of basis functions and models the projection coefficient vectors within each cluster as lying in a low-dimensional subspace. A parametric model is then applied to these coefficient vectors for clustering. funHDDC falls under the third category (lines 24-26) and is probabilistic in nature.
> - funclust [2]: This method also employs a parametric model but uses functional PCA scores (instead of general basis coefficients) as inputs for clustering. It falls under the first category and is also probabilistic.
> - FADPclust [3]: This method applies the density peak clustering algorithm directly to the pairwise distance matrix of functional data, making it a deterministic method in the second category.
>
> [1] Bouveyron and Jacques. 2011. Model-based clustering of time series in group-specific functional subspaces. Advances in Data Analysis and Classification.
>
> [2] Jacques and Preda. 2013. Funclust: A curves clustering method using functional random variables density approximation. Neurocomputing.
>
> [3] Wang and Xu 2015. Fast clustering using adaptive density peak detection. Statistical Methods in Medical Research.
>
> **Response to Q2:** All baseline methods (including FNN, FAE and VANO) require the number of clusters to be specified as an input. In contrast, FAEclust automatically determines the number of clusters in a fully data-driven manner using an internal validation index.
>
> For all baseline methods, we run each method with the number of clusters ranging from 2 to 10, ensuring that the true number of clusters is included in this range. We then report the best clustering performance based on ARI and AMI.
>
> For FAEclust, the number of clusters is adaptively determined within each forward phase of the joint training and clustering framework. Please refer to our first response to Reviewer CGH6 for a detailed explanation of this procedure.
>
> **Response  to Q3 and Limitation 1:** Thanks for the suggestion. We will give more explanation after Theorem 1 in the revision.
> 1.  Why Theorem 1 matters: **(a)** When a decoder cannot faithfully reconstruct every possible latent code, the encoder quickly discovers which regions of the latent space will lead to plausible outputs and which will not. In effect, the encoder “steers clear” of those latent directions where the decoder’s mapping breaks down. As a result, the geometry of the learned representation is warped: some dimensions or combinations of dimensions are never explored, because they would produce meaningless or distorted reconstructions. Consequently, any clustering performed on this latent representation yields groupings that conform to the decoder’s permissible manifold rather than to the data’s intrinsic structure.  **(b)** Additionally, limited decoder capacity leads to consistent failures on certain patterns or classes. The network compensates by balancing reconstruction loss against clustering objectives, resulting in unpredictable trade-offs that undermine both reconstruction quality and cluster integrity. **(c)** Finally, even small differences between the initial clusters (after pre-training) and the final clusters (after fine-tuning) become magnified by a weak decoder, making the learned clusters fragile to noise. Theorem 1 guarantees that our decoder has the universal-approximation property, so none of these pathologies can arise.
> 2. Connection to classic universal approximation theorem: This was briefly explained in lines 399-405. While the universal-approximation property of finite-dimensional neural networks is well understood, far less is known about infinite-dimensional mappings ([6], [11]). Unlike existing functional networks that rely on integral kernels, our framework uses univariate functions as weights. This key distinction requires a bespoke universality proof. In our paper, we not only establish Euclidean universal approximation but also extend it to patchwise approximation on compact, connected Riemannian manifolds, substantially strengthening and generalizing the classical result.
> 3. The full construction of the readout map $\rho$ appears later in lines 479-485. In the revised manuscript, we will add a concise explanation of $\rho$ directly in the main text.
>
> **Response to Q4 and Limitation 2:** Phase variation (a.k.a., time warping) in functional data refers to variability in the timing of key features across curves. Addressing phase variation in functional data is crucial because it distinguishes true shape differences from misalignments in time or space. In most functional data studies, curve alignment is treated as a separate pre-processing step, independent of subsequent analysis. As noted in [34] (Section 7), only a small number of works explicitly integrate phase variation handling within clustering, highlighting the challenge of this problem.
>
> Unlike classical symmetry groups such as the Euclidean group E(2), the group of warping functions (acting on functions via composition) is nonlinear and does not admit a standard (left) group representation. This nonlinear structure means we cannot easily design network layers that are equivariant or invariant to phase variation in the same way as for transformations such as rotation. In our current work, we handle phase variation through the clustering objective that incorporates a shape-informed elastic metric, which captures phase-amplitude separation. Developing neural network architectures explicitly designed to identify and disentangle phase variation remains an important future direction.
>
> In Table 1, most datasets there do not exhibit phase variation. To investigate the effect of ignoring phase variation, we applied the L2-variant of FAEclust (replacing the elastic distance in Eq. (6) with the standard L2 distance) across the 12 simulation scenarios. The clustering results are shown below. For comparison with other baseline methods, please refer to the second table in our response to Reviewer WLrd. These results highlight the advantages of using a shape-informed metric for clustering problems where phase variation is a nuisance.
>
> | |Warped | | Warped| |
> |:--|:--|:--|:--|:--|
> |Scenario|Elastic||$L^2$||
> ||AMI|ARI|AMI|ARI|
> |A|0.751|0.724|0.582|0.465|
> ||0.016|0.009|0.021|0.038|
> |B|0.405|0.383|0.009|0.002|
> ||0.092|0.114|0.004|0.001|
> |C|0.665|0.597|0.574|0.460|
> ||0.109|0.073|0.042|0.059|
> |D|0.559|0.526|0.274|0.216|
> ||0.088|0.062|0.011|0.019|
> |E|0.851|0.789|0.717|0.570|
> ||0.038|0.026|0.034|0.028|
> |F|0.839|0.781|0.735|0.613|
> ||0.032|0.045|0.024|0.036|
> |G|0.585|0.487|0.086|0.051|
> ||0.103|0.158|0.009|0.012|
> |H|0.997|0.991|0.734|0.672|
> ||0.011|0.006|0.037|0.029|
> |I|0.968|0.955|0.733|0.591|
> ||0.009|0.008|0.022|0.045|
> |J|0.142|0.116|0.003|0.002|
> ||0.092|0.089|0.001|0.002|
> |K|0.876|0.824|0.163|0.094|
> ||0.032|0.024|0.038|0.037|
> |L|0.359|0.370|0.419|0.361|
> ||0.108|0.123|0.064|0.080|
> ||||||
>
> **Response to Limitation 3:** Thank you for the suggestion. In Table 1, most datasets are small (around 50 curves) to medium (around 200 curves) in size. For example, BirdChicken has 40 curves, Rock has 70, and BM has 80. Bootstrapping is less suitable for unsupervised learning because clustering depends on global data structure and lacks ground truth labels to anchor resamples. Resampling can distort the underlying relationships, especially in small datasets, and lead to unstable or non-comparable cluster solutions. To assess the robustness of FAEclust, we instead ran FAEclust 20 times per dataset and report the mean and standard deviation of AMI and ARI in the tables above (in responses to Reviewer CGH6 and to Reviewer WLrd). The observed variation arises from the stochastic nature of Bayesian optimization and the mini-batch gradient descent algorithm.
>
> For the 12 simulation scenarios, we generated 100 independent datasets for each scenario and report the mean (top row) and standard deviation (bottom row) of ARI and AMI, thereby quantifying uncertainty due to both data variability and algorithmic randomness.

---

> ### Comment · Reviewer_5zSQ · 2025-08-01
>
> Thanks for the detailed rebuttal, it clarifies my original questions. In general I think the idea of clustering multivariate functional data using autoencoder while taking phase variation into consideration is interesting. Also, the authors successfully illustrate that the proposed method performs better than benchmarks. I will raise my overall rating.

---

### Official Review · Reviewer_WLrd · 2025-06-24

**Clarity:** 2
**Significance:** 2
**Originality:** 3
**Rating:** 4
**Confidence:** 4

**Summary:**

The authors propose FAEclust, a clustering algorithm for functional data, based on a functional autoencoder. It jointly learns a  nonlinear, shape-aware embeddings and performs convex clustering for multi-dimensional functional data. Across many Euclidean and five manifold datasets FAEclust consistently outperforms classic functional-data baselines, though performance drops on extremely low-signal manifold cases.

**Questions:**

-Please answer the question regarding the limitation below. **And**
As noted, the experimental evaluation would be more convincing if it included methods [9] and [23], which are closely related and represent strong baselines. Is it possible to include them in experiments?

If the authors adequately address the points raised above—particularly by discussing the observed limitations and including the suggested baselines—I would be inclined to raise my score. These additions would significantly strengthen the paper’s empirical foundation and overall clarity.

**Ethical Concerns:**

["NO or VERY MINOR ethics concerns only"]

**Final Justification:**

In their rebuttal, the authors provided detailed results for the previously omitted methods, thereby enhancing the overall quality of the paper.

**Limitations:**

The paper demonstrates commendable breadth by evaluating on a diverse set of datasets—both Euclidean and manifold—and consistently outperforms selected baselines. However, key insights into why the method performs better, and under what conditions it fails, are either missing or only superficially addressed. This weakens the overall empirical evidence chain. For example, on many manifold datasets (Table 2), the clustering performance remains below 0.4, which raises concerns about the method's robustness in such settings. Could the authors provide an explanation for this behavior? It would also strengthen the paper to explicitly acknowledge this limitation in the discussion section.

**Paper Formatting Concerns:**

-

**Quality:**

2

**Strengths And Weaknesses:**

The paper proposes a new algorithm and strong results over some competitors (funHDDC, funclust, FADP). Unfortunately, the paper fails in giving the comprehensive details of these related works, does not even cite all of them. Instead, a few other related works are cited, e.g. [9] and [23] (see their paper) which appear to be more related as they apply also autoencoders to functional data. They exclude them from experimentation by the hint "substantial computational challenges". This is too general and does not convince me.

While I believe that the algorithm is indeed new, the paper needs to discuss the novelty in the light of related work, not just pointing out "my algorithm yields better numbers than something..."
Overall many components are known from the paper. The paper combines them in unique way therefore the novelty is medium.


The results in Table 1 are promising and suggest strong overall performance. However, the outcomes in Table 2 warrant more careful consideration. In particular, the consistently low performance on several manifold datasets calls for a deeper explanation. Clarifying the underlying factors contributing to these results—such as dataset characteristics, model limitations, or failure modes—would provide a more balanced and informative assessment of the method's strengths and weaknesses.

As noted, the experimental evaluation would be more convincing if it included methods [9] and [23], which are closely related and represent strong baselines. Including these would provide a more comprehensive comparison and help better contextualize the reported performance gains.

---

> ### Author Rebuttal · Authors · 2025-07-25
>
> Thank you for these valuable suggestions. We have explained the novelty of our work in our response to Reviewer 9SuC. Before presenting additional comparative studies, we would like to clarify our choice of benchmarks: We compared FAEclust to every method surveyed in [34] that (a) is explicitly designed for clustering multi-dimensional functional data, and (b) provides publicly available code. While the approaches in [7] and [9] address similar problems, their implementations are not accessible. We did reach out to the authors of the two works when preparing the benchmark study; however, both informed us that they were unable to share their implementations. We excluded [23] because its autoencoder processes input and output as plain vectors of function evaluations, hence not fundamentally a functional network.
>
> To ensure a fair and comprehensive comparison, we re-implemented the methods from [7] **and** [9] ourselves and included them in our comparative study, along with [23]. The network architecture of [7] follows Figure 1 in that work, while the architecture of [9] is based on Algorithm 1. For each method, we used Bayesian optimization for hyperparameter tuning.
>
> In the table below, we compare the four methods across the 17 real datasets, with each model run 20 times per dataset. The reported variation is due to the stochastic nature of Bayesian optimization and the network training process. FAEclust consistently outperforms the other three methods on the majority of datasets.
>
> |Datasets|&emsp;&emsp;&emsp;&emsp;&emsp;&emsp;FAEclust|&emsp;&emsp;&emsp;&emsp;&emsp;FNN[7]+K-means|&emsp;&emsp;&emsp;&emsp;&emsp;FAE[9]+K-means|&emsp;&emsp;&emsp;  VANO[23]+K-means|
> |--|--|--|--|--|
> | |&emsp;&emsp;&emsp;**AMI**&emsp;&emsp;&emsp;&emsp;&emsp;&emsp;**ARI**|&emsp;&emsp;&emsp; **AMI**&emsp;&emsp;&emsp;&emsp;&emsp;&emsp;**ARI**|&emsp;&emsp;&emsp;**AMI**&emsp;&emsp;&emsp;&emsp;&emsp;&emsp;**ARI**|&emsp;&emsp;&emsp;**AMI**&emsp;&emsp;&emsp;&emsp;&emsp;&emsp;**ARI**|
> |BirdChicken|&emsp;$0.329\\pm0.037$&emsp;$0.260\\pm0.052$|&emsp;$0.282\\pm0.026$&emsp;$0.219\\pm0.032$|&emsp;$0.266\\pm0.023$&emsp;$0.225\\pm0.033$|&emsp;$0.287\\pm0.030$&emsp;$0.164\\pm0.042$|
> |CBF|&emsp;$0.719\\pm0.043$&emsp;$0.689\\pm0.054$|&emsp;$0.367\\pm0.013$&emsp;$0.338\\pm0.020$|&emsp;$0.361\\pm0.015$&emsp;$0.343\\pm0.025$|&emsp;$0.723\\pm0.032$&emsp;$0.653\\pm0.040$|
> |Chinatown|&emsp;$0.319\\pm0.024$&emsp;$0.252\\pm0.020$|&emsp;$0.264\\pm0.035$&emsp;$0.198\\pm0.087$|&emsp;$0.132\\pm0.021$&emsp;$0.116\\pm0.027$|&emsp;$0.313\\pm0.038$&emsp;$0.239\\pm0.057$|
> |DSR|&emsp;$0.881\\pm0.029$&emsp;$0.865\\pm0.041$|&emsp;$0.688\\pm0.095$&emsp;$0.579\\pm0.112$|&emsp;$0.636\\pm0.030$&emsp;$0.462\\pm0.052$|&emsp;$0.704\\pm0.051$&emsp;$0.637\\pm0.063$|
> |ECG200|&emsp;$0.240\\pm0.035$&emsp;$0.160\\pm0.065$|&emsp;$0.230\\pm0.039$&emsp;$0.134\\pm0.032$|&emsp;$0.215\\pm0.035$&emsp;$0.122\\pm0.041$|&emsp;$0.219\\pm0.028$&emsp;$0.125\\pm0.023$|
> |Fungi|&emsp;$0.926\\pm0.019$&emsp;$0.847\\pm0.048$|&emsp;$0.261\\pm0.018$&emsp;$0.148\\pm0.013$|&emsp;$0.604\\pm0.025$&emsp;$0.475\\pm0.030$|&emsp;$0.821\\pm0.027$&emsp;$0.687\\pm0.042$|
> |Plane|&emsp;$0.904\\pm0.016$&emsp;$0.826\\pm0.039$|&emsp;$0.842\\pm0.031$&emsp;$0.787\\pm0.036$|&emsp;$0.832\\pm0.013$&emsp;$0.752\\pm0.016$|&emsp;$0.855\\pm0.029$&emsp;$0.702\\pm0.041$|
> |Rock|&emsp;$0.455\\pm0.023$&emsp;$0.309\\pm0.026$|&emsp;$0.198\\pm0.037$&emsp;$0.160\\pm0.033$|&emsp;$0.104\\pm0.028$&emsp;$0.116\\pm0.021$|&emsp;$0.354\\pm0.027$&emsp;$0.256\\pm0.032$|
> |Symbols|&emsp;$0.839\\pm0.031$&emsp;$0.674\\pm0.039$|&emsp;$0.748\\pm0.005$&emsp;$0.631\\pm0.026$|&emsp;$0.793\\pm0.008$&emsp;$0.645\\pm0.010$|&emsp;$0.794\\pm0.020$&emsp;$0.662\\pm0.041$|
> |Blink|&emsp;$0.637\\pm0.038$&emsp;$0.572\\pm0.043$|&emsp;$0.483\\pm0.039$&emsp;$0.430\\pm0.048$|&emsp;$0.510\\pm0.031$&emsp;$0.455\\pm0.014$|&emsp;$0.497\\pm0.045$&emsp;$0.419\\pm0.051$|
> |BM|&emsp;$0.542\\pm0.037$&emsp;$0.493\\pm0.046$|&emsp;$0.393\\pm0.044$&emsp;$0.326\\pm0.031$|&emsp;$0.671\\pm0.061$&emsp;$0.655\\pm0.050$|&emsp;$0.596\\pm0.042$&emsp;$0.530\\pm0.058$|
> |EOS|&emsp;$0.261\\pm0.014$&emsp;$0.206\\pm0.019$|&emsp;$0.161\\pm0.036$&emsp;$0.068\\pm0.049$|&emsp;$0.206\\pm0.024$&emsp;$0.163\\pm0.020$|&emsp;$0.179\\pm0.036$&emsp;$0.136\\pm0.056$|
> |Epilepsy|&emsp;$0.464\\pm0.039$&emsp;$0.288\\pm0.034$|&emsp;$0.290\\pm0.027$&emsp;$0.181\\pm0.013$|&emsp;$0.169\\pm0.033$&emsp;$0.122\\pm0.025$|&emsp;$0.176\\pm0.023$&emsp;$0.118\\pm0.018$|
> |ERing|&emsp;$0.657\\pm0.047$&emsp;$0.582\\pm0.053$|&emsp;$0.662\\pm0.052$&emsp;$0.593\\pm0.011$|&emsp;$0.762\\pm0.027$&emsp;$0.736\\pm0.036$|&emsp;$0.734\\pm0.040$&emsp;$0.684\\pm0.063$|
> |FM|&emsp;$0.261\\pm0.034$&emsp;$0.180\\pm0.051$|&emsp;$0.187\\pm0.035$&emsp;$0.094\\pm0.047$|&emsp;$0.159\\pm0.016$&emsp;$0.117\\pm0.023$|&emsp;$0.192\\pm0.029$&emsp;$0.086\\pm0.058$|
> |JV|&emsp;$0.883\\pm0.016$&emsp;$0.870\\pm0.020$|&emsp;$0.258\\pm0.072$&emsp;$0.098\\pm0.032$|&emsp;$0.857\\pm0.006$&emsp;$0.841\\pm0.008$|&emsp;$0.892\\pm0.006$&emsp;$0.883\\pm0.009$|
> |SWJ|&emsp;$0.322\\pm0.063$&emsp;$0.257\\pm0.051$|&emsp;$0.146\\pm0.039$&emsp;$0.127\\pm0.018$|&emsp;$0.154\\pm0.019$&emsp;$0.132\\pm0.030$|&emsp;$0.337\\pm0.068$&emsp;$0.286\\pm0.055$|
> | | | | | |
>
> In the table below, we compare the four methods on the 12 simulated time-warped datasets. We generate 100 datasets for each scenario. FAEclust demonstrates robust and consistently superior performance across most scenarios. The drop of performance in Scenario L has been explained in Appendix E.3.
>
> ||Original|Warped|Warped|Warped|Warped|
> |--|--|--|--|--|--|
> |Scenario|FAEclust|FAEclust|FNN|FAE|VANO|
> | |AMI ARI|AMI ARI|AMI ARI|AMI ARI|AMI ARI|
> |A|0.756 0.725|0.751 0.724|0.443 0.384|0.437 0.334|0.616 0.560|
> | |0.010 0.006|0.016 0.009|0.062 0.068|0.017 0.013|0.034 0.034|
> |B|0.426 0.446|0.405 0.383|0.003 0.004|0.002 0.003|0.004 0.005|
> | |0.069 0.084|0.092 0.114|0.008 0.011|0.006 0.009|0.010 0.013|
> |C|0.671 0.608|0.665 0.597|0.499 0.236|0.450 0.416|0.561 0.484|
> | |0.096 0.081|0.109 0.073|0.004 0.007|0.018 0.022|0.031 0.048|
> |D|0.573 0.529|0.559 0.526|0.041 0.038|0.042 0.035|0.294 0.257|
> | |0.094 0.075|0.088 0.062|0.019 0.021|0.018 0.013|0.038 0.039|
> |E|0.846 0.776|0.851 0.789|0.763 0.773|0.743 0.696|0.728 0.692|
> | |0.054 0.051|0.038 0.026|0.035 0.029|0.015 0.020|0.123 0.171|
> |F|0.864 0.793|0.839 0.781|0.309 0.185|0.477 0.358|0.658 0.548|
> | |0.046 0.039|0.032 0.045|0.035 0.034|0.035 0.046|0.047 0.050|
> |G|0.579 0.478|0.585 0.487|0.285 0.222|0.284 0.226|0.216 0.170|
> | |0.127 0.142|0.103 0.158|0.034 0.031|0.047 0.038|0.055 0.041|
> |H|0.998 0.993|0.997 0.991|0.629 0.542|0.723 0.709|0.858 0.848|
> | |0.008 0.005|0.011 0.006|0.040 0.022|0.026 0.032|0.055 0.068|
> |I|0.971 0.960|0.968 0.955|0.632 0.550|0.658 0.552|0.687 0.652|
> | |0.018 0.017|0.009 0.008|0.034 0.022|0.019 0.090|0.049 0.062|
> |J|0.151 0.080|0.142 0.116|0.001 0.001|0.000 0.001|0.002 0.002|
> | |0.076 0.053|0.092 0.089|0.004 0.005|0.005 0.007|0.003 0.003|
> |K|0.873 0.825|0.876 0.824|0.082 0.106|0.083 0.110|0.271 0.260|
> | |0.022 0.015|0.032 0.024|0.082 0.103|0.084 0.110|0.049 0.050|
> |L|0.704 0.729|0.359 0.370|0.656 0.678|0.636 0.702|0.632 0.640|
> | |0.087 0.064|0.108 0.123|0.048 0.042|0.035 0.038|0.066 0.086|
> | | | | | | |
>
> The lower performance in Table 2 is due to the narrow architecture of FAEclust. We should have provided more detail regarding the architecture choices of FAEclust. In all experiments—both real and simulated—we fixed the architecture to a seven-layer structure: one functional layer in the encoder, three functional layers in the decoder, and a three-layer MLP in between. The third layer in FAEclust serves as the bottleneck, producing the latent embedding. Please refer to our response to Reviewer CGH6 for an explanation of this choice. The maximum number of nodes allowed per layer was constrained to {32, 16, 8, 16, 32, 32, 32}. For the five datasets in Table 2, the final architectures were {16, 8, 2, 8, 16, 16, 16} for Hypersphere and Lorenz, and {32, 16, 8, 16, 32, 32, 32} for Hyperbolic, Swiss Roll, and Pendulum. Even under this restricted architecture, FAEclust consistently outperforms all benchmark methods.
>
> While a small network may guard against overfitting with only 100-300 samples, it underfits when tasked with capturing intricate, non‑linear patterns. If we were to increase the model capacity, FAEclust could achieve even better performance. The below table shows the new clustering results, where we expand the width-search space to {64, 32, 16, 32, 64, 64, 64}. These results confirm that FAEclust can achieve better performance when given additional capacity, especially on more complex datasets.
>
> |Datasets|FAEclust|||FNN||FAE||VANO||
> |:--|:--|:--|:--|:--|:--|:--|:--|:--|:--|
> | |layer width|AMI|ARI|AMI|ARI|AMI|ARI|AMI|ARI|
> |Hypersphere|{64, 32, 16, 32, 64, 32, 32}|0.649|0.519|0.016|0.023|0.050|0.037|0.343|0.327|
> |Hyperbolic|{64, 32, 16, 32, 32, 32, 32}|0.842|0.814|0.001|0.003|0.043|0.060|0.028|0.039|
> |Swiss Roll|{64, 32, 16, 32, 64, 64, 64}|0.459|0.376|0.014|0.018|0.120|0.106|0.104|0.068|
> |Lorenz|{32, 16, 16, 16, 32, 32, 32}|0.488|0.440|0.023|0.028|0.154|0.106|0.229|0.180|
> |Pendulum|{32, 16, 16, 16, 16, 16, 32}|1|1|0.254|0.195|0.781|0.621|0.958|0.941|
> | | | | | | | | | | |

---

> > ### Comment · Reviewer_WLrd · 2025-08-02
> >
> > Thank you for the detailed rebuttal and the new baseline comparisons. I will raise my overall score.

---

### Official Review · Reviewer_9SuC · 2025-06-27

**Clarity:** 2
**Significance:** 2
**Originality:** 2
**Rating:** 4
**Confidence:** 4

**Summary:**

The paper presents an autoencoder-based approach (FAEclust) to clustering multidimensional functional data. It is based on embeddings created by training a functional autoencoder minimizing a combined loss related to reconstruction error and clustering error. Moreover, an additional clustering objective (“shape-informed” objective) is optimized that takes into account the pairwise similarity of functional input data. Weight functions are represented as weighted linear sums of basis functions (qubic B-splines).

**Questions:**

Q1. It is not easy to judge the novelty of the proposed approach. The novel aspects with respect to its competitors are not sufficiently highlighted.
Q2. The use of two clustering objective functions creates some confusion. The notation U is used in both L_c and L_s, however their dimensions are different. Why not integrating the regularization term of (5) into L_c, instead of performing separate optimizations?
Q3. A pseudocode is needed to describe in detail the steps of the full method. Figure 2 is by no means sufficient. The algorithm in lines 515-530 refer to a loss function L, but it is not clear what L represents.
Q4. How do the results change if the cluster update is omitted?
Q5. There too many hyperparameters that need to be set in order to apply the method (including the number m of basis functions). It is not clear how all those parameters are set. If they are set taking into account the ground truth labels, then this procedure is not acceptable in a clustering setting. Typically, default values should be provided for all hyperparameters.
Q6. The experiments with competitors are made using their default values, while for the proposed method a search for optimal hyperparameter values is conducted.
Q7. In several datasets with AMI/ARI values are small (less than 0.35). This means that the ground truth clustering structure is not recovered, thus comparison is meaningless.
Q8. There is no information on whether the ground truth number of clusters is recovered or not. It is just mentioned that an internal validation metric is used.

**Ethical Concerns:**

["NO or VERY MINOR ethics concerns only"]

**Final Justification:**

Based on the clarifications provided, the additional experimental results and the positive opinion of other reviewers I have increased my score.

**Limitations:**

The main limitations of the method concern hyperparameter tuning and execution time.

**Paper Formatting Concerns:**

No concern.

**Quality:**

2

**Strengths And Weaknesses:**

Strengths
S1. The use of the shape informed objective introduces some novelty in the method.
S2. Theoretical results are provided on the approximation capability of the functional decoder.
S3. Better results have been obtained compared to other functional clustering approaches.

Weaknesses
W1. There are several concerns related to novelty, methodology, presentation and clarity (see questions section).
W2. The method involves too many hyperparameters. It is mentioned that they are determined “through Bayesian optimization over a predefined search space”. It is not clear if such an optimization requires the ground truth cluster labels or not.

---

> ### Author Rebuttal · Authors · 2025-07-24
>
> **Q1:  "It is not easy to judge the novelty of the proposed approach."**
>
> We apologize for the brevity in our original submission; space constraints limited how much detail we could include. The main contributions of the work are as follows.
>
> 1. FAEclust is the first autoencoder architecture built specifically to multi-dimensional functional data *cluster analysis*. By using functional weights, it operates directly on functions rather than pointwise values, and its deep structure automatically captures complex dependencies among the component random functions.
> 2. We establish that our decoder—which maps from a finite-dimensional space into an infinite‐dimensional function space—is a universal approximator in the Euclidean sense, and furthermore achieves patchwise universal approximation on any compact, connected Riemannian manifold. (This theoretical guarantee ensures the encoder never “avoid” regions of latent space for fear that the decoder cannot reconstruct them, a point we’ve further detailed in our response to Reviewer 5zSQ.)
> 3. To address phase variation in functional datasets, we introduce a clustering objective that simultaneously captures shape information and remains separable across dimensions. We devise a fully data-driven method for selecting the neighborhood parameter m, and deliver an O(nlog(n)) optimization algorithm.
> 4. FAEclust is available as a fully documented Python package (anonymous GitHub link in the paper), with sensible default hyperparameter settings.  Unlike related methods without released code, our package enables practitioners to apply, reproduce, and extend FAEclust immediately.
>
>
> **Q2:  "The use of two clustering objective functions creates some confusion. The notation U is used in both L_c and L_s, ... Why not integrating the regularization term of (5) into Lc, instead of performing separate optimizations?"**
>
> Thank you for catching that ambiguity. We will rename $\pmb{u}_k$ in $\mathcal{L}_c$ to avoid confusion.
>
> $\mathcal{L}_c$  is a regularization term in our training objective; it helps shape the latent space for clustering but is not itself a clustering criterion. Both  $\mathcal{L}_c$ and  $\mathcal{L}_s$ include a first term that measures within-cluster dispersion. Their key difference lies in the second term. In  $\mathcal{L}_c$, the second term measures between-cluster dispersion, so adding  $\mathcal{L}_c$ to the training objective encourages clusters to spread apart in the embedding space. In $\mathcal{L}_s$, the second term captures shape information in the original functional data. Therefore, using $\mathcal{L}_c$ in the loss prevents the latent representations from collapsing, a problem that can arise if one replaces  $\mathcal{L}_c$ with $\mathcal{L}_s$ in the network-training objective.
>
> Why performing separate optimizations: network training and clustering serve distinct purposes and involve different objective functions. Once the network parameters (and hence the embeddings) are updated, the number of clusters and cluster centroids need be updated. In other words, both the number of clusters and the cluster centroids are unknown and dynamically determined. As such, they cannot be directly incorporated into the network training objective, which requires a fixed structure.
>
> **Q3:  "A pseudocode is needed ... The algorithm in lines 515-530 refer to a loss function L, but it is not clear what L represents."**
>
> In the backward network-training phase (see Figure 2),  we use stochastic gradient descent (Appendix D) to minimize the training objective function. In the forward cluster-update phase, we use the homotopy algorithm (Section 4.2) to minimize the clustering objective function. In the Python package (anonymous GitHub link in the paper), we explained the modular pipeline of the full method.
>
> Appendix D is for explaining and justifying the optimization algorithm for training the network, and hence the loss function $\mathcal{L}$ there is the training objective function ($\mathcal{L}=\mathcal{L}_r+\lambda_w\mathcal{L}_w+\lambda_c\mathcal{L}_c$ in lines 141 & 142).
>
> A high-level logic of the  path-following homotopy algorithm is below:
>
> **Input**: latent embeddings $\pmb{X}$, pairwise similarities $s(\pmb{y}_i, \pmb{y}_j)$, and $\lambda=0$ (or a warm start).
>
> **While** number of clusters > 2:
>
> - Calculate the next breakpoint $\lambda_{K-1}$ where a pair of centroids fuse (line 302).
> - Merge the corresponding clusters.
>
> **Return**: breakpoints {$\lambda_{n-1}$, ..., $\lambda_2$} and corresponding hierarchical clusterings.
>
> We will integrate the above logic, Appendix D, and Figure 2 to provide a clearer explanation of the full method. Thanks for the suggestion.
>
> **Q4:  "How do the results change if the cluster update is omitted?”**
>
> Our training process follows the usual pattern in “deep clustering” (as explained in lines 152-155): (1) pre-train an autoencoder on penalized reconstruction-loss only; (2) initialize clusters; (3) joint fine-tuning. If cluster update is omitted, then the final clusters are just the initial clusters from step (2). The pre-training only minimizes the reconstruction loss, and therefore the latent representations are agnostic to clustering structure (e.g., shape information). Iterative updating aligns latent representations with the clustering objective, leading to better-defined, more meaningful clusters.
>
> **Q5 and Weakness 2: "There are too many hyperparameters. … If they are set taking into account the ground truth cluster labels, then this procedure is not acceptable in a clustering setting.”**
>
> We will provide more information in Appendix E.1. As explained in lines 549-588, FAEclust shares hyperparameters with standard autoencoders. The additional hyperparameters are related to the functional nature of the network and the two penalty parameters ($\lambda_c$ and $\lambda_w$). Note that $\lambda$ in Eq. (5) is not a hyperparameter; please see our first response to Reviewer CGH6 for further clarification. Tuning hyperparameters using Bayesian optimization is a standard and well-established approach for machine learning hyperparameter tuning (see, e.g., [1]). There’s no need for ground-truth cluster labels. Users simply define each parameter’s search domain. FAEclust is available as a fully documented Python package (anonymous GitHub link in the paper). Its README lists sensible default values for every hyperparameter.
>
> [1]: Turner, R. et al. “Bayesian Optimization is Superior to Random Search for Machine Learning Hyperparameter Tuning: Analysis of the Black-Box Optimization Challenge 2020”. Proceedings of the NeurIPS 2020 Competition and Demonstration Track, PMLR 133:3-26, 2021.
>
> **Q6:  "The experiments with competitors are made using their default values, while for the proposed method a search for optimal hyperparameter values is conducted."**
>
> We would like to clarify that for all baseline methods, we configured each algorithm to achieve its best possible performance, rather than simply using default settings.
>
> - funHDDC: We evaluated all six supported model types (line 538) and reported the one that gave the best performance for each dataset. Other parameters were related to algorithm convergence.
> - funclust: The parameter thd is the significance level in a hypothesis test, and setting thd=0.05 is a well-accepted choice. The parameters increaseDimension = FALSE and hard = FALSE were also tested and selected based on performance. Other arguments were kept at default as they pertain only to algorithmic stability or convergence.
> - FADPclust: We set proportion = NULL, which triggers the algorithm's internal grid search over 10 candidate values, as recommended by the authors.
> - Across all baseline methods, we conducted a grid search over the number of clusters and reported the result that achieved the best clustering performance. In contrast, FAEclust determines the number of clusters in a fully data-driven manner, as explained in our response to Reviewer CGH6.
>
> **Q7:  "In several datasets with AMI/ARI values are small (less than 0.35). This means that the ground truth clustering structure is not recovered, thus comparison is meaningless."**
>
> Thank you for the comment. We should have clarified that FAEclust was evaluated using a fixed architecture across all experiments: 7 layers in total (1 encoder + 3 decoder functional layers, and a 3-layer MLP), with maximum layer widths of {32, 16, 8, 16, 32, 32, 32}. Even with this conservative setup, FAEclust consistently outperformed all baseline methods. The key message we aim to convey is that FAEclust is substantially more effective than existing approaches.
>
> If we were to increase the model capacity, FAEclust could achieve even better performance. For example, on the challenging hypersphere dataset, expanding the search space to {64, 32, 16, 32, 64, 64, 64} enables Bayesian optimization to find an architecture {64, 32, 16, 32, 64, 32, 32}, which achieves ARI = 0.5194 and AMI = 0.6489. Moreover, increasing the sample size from 100 to 500 yields ARI = 1 and AMI = 1, a perfect recovery of the true clustering. Please see our response to Reviewer WLrd for further details and additional supporting experiments.
>
> **Q8: "There is no information on whether the ground truth number of clusters is recovered or not. It is just mentioned that an internal validation metric is used.''**
>
> We use an internal validation metric to determine the number K of clusters in each forward phase, much like using classification accuracy in a supervised setting; please see our first response to Reviewer CGH6 for further clarification. To evaluate how well FAEclust recovers the true cluster structure, in our experiments, we reported AMI and ARI, both of which are computed against the ground-truth labels. We will include the number of identified clusters in our revised version.
>
> We hope our responses have addressed the reviewer’s concerns. We would be pleased to provide any further clarification during the discussion period.

---

> > ### Comment · Reviewer_9SuC · 2025-08-01
> > **questions on author rebuttal**
> >
> > Thank you for your informative reply. I still have some concerns:
> > Q5 reply: Since you are not  using ground truth labels, could you please clarify which is the score that you optimize in the Bayesian search for optimum hyperparameters?
> > Q8 reply: I was expecting that you would include in the rebuttal information on the estimated number of clusters K. It is typical to provide both (AMI, ARI)  values and the value of K when evaluating clustering algorithms with unknown K.

---

> ### Author Response · Authors · 2025-08-01
> **More clarification on Q5 and Q8**
>
> Response Q5 comment: The objective function in the Bayesian optimization problem is still the integrated objective function $\mathcal{L}=\mathcal{L}_r+\lambda_w\mathcal{L}_w+\lambda_c\mathcal{L}_c$. However, in the context of Bayesian optimization, the variables being optimized are the hyperparameters (e.g., number of neurons per layer, dropout rate, etc.), rather than the model weights. In our implementation, we utilized the Optuna package to perform the hyperparameter tuning task.
>
> Response Q8 comment: We report below the number of clusters identified by each clustering method across the 17 real datasets. The numbers in the parentheses are the true number of clusters.
>
> | Dataset           | FAEclust | funHDDC | funclust | FADP1 | FADP2 | FNN | FAE | VANO |
> |------------------|----------|---------|----------|-------|-------|-----|-----|------|
> | BirdChicken (2)  | 2        | 3       | 4        | 2     | 3     | 2   | 2   | 2    |
> | CBF (3)          | 3        | 3       | 2        | 2     | 3     | 2   | 2   | 3    |
> | Chinatown (2)    | 2        | 4       | 2        | 2     | 2     | 2   | 2   | 2    |
> | DSR (4)          | 6        | 4       | 3        | 3     | 2     | 4   | 3   | 4    |
> | ECG200 (2)       | 2        | 3       | 2        | 2     | 2     | 2   | 2   | 2    |
> | Fungi (18)       | 16       | 8       | 2        | 2     | 5     | 7   | 8   | 12   |
> | Plane (7)        | 7        | 5       | 2        | 4     | 3     | 6   | 6   | 6    |
> | Rock (4)         | 3        | 4       | 7        | 2     | 2     | 2   | 2   | 2    |
> | Symbols (6)      | 4        | 4       | 2        | 3     | 4     | 4   | 4   | 4    |
> | Blink (2)        | 2        | 2       | 2        | 2     | 2     | 2   | 2   | 2    |
> | BM (4)           | 3        | 2       | 5        | 2     | 2     | 3   | 3   | 3    |
> | EOS (2)          | 2        | 2       | 2        | 2     | 2     | 2   | 3   | 2    |
> | Epilepsy (4)     | 4        | 8       | 2        | 2     | 3     | 3   | 3   | 3    |
> | ERing (6)        | 3        | 4       | 2        | 2     | 4     | 3   | 4   | 4    |
> | FM (2)           | 4        | 2       | 2        | 2     | 6     | 3   | 3   | 4    |
> | JV (9)           | 8        | 8       | 2        | 2     | 4     | 2   | 8   | 8    |
> | SWJ (3)          | 2  | 2 | 2        | 2     | 2     | 2   | 2   | 2    |
> |  |   |   |  |   |   |   |  | |
>
> Below are the number of clusters identified by each clustering method across the five manifold datasets.
>
> | Dataset           | FAEclust |FNN | FAE | VANO | funHDDC | funclust | FADP1 | FADP2 |
> |------------------|----------|---------|----------|-------|-------|-----|-----|------|
> | Hypersphere (2)  | 2        | 6       | 4        | 2     | 5     | 2   | 2   | 4    |
> | Hyperbolic (2)     | 2        | 4       | 3        | 2     | 6     | 2   | 2   | 3    |
> | Swiss Roll (4)     | 4        | 2       | 2        | 3     | 3     | 2   | 4   | 2    |
> | Lorenz (3)           | 4        | 4       | 4        | 2     | 3     | 3   | 2   | 3    |
> | Pendulum (4)      | 4        | 2       | 4        | 4     | 4     | 3   | 4   | 4    |
> |  || | |  |  | |  | |

---

> > ### Comment · Reviewer_9SuC · 2025-08-04
> > **results on number of clusters**
> >
> > Thank you for the information on the number of clusters. Since the results are integers, are those values obtained from a single run on each dataset?

---

> > > ### Author Response · Authors · 2025-08-04
> > > **Response to results on number of cluters.**
> > >
> > > Each integer indicates the number of clusters that appeared most often across 20 repetitions. We will report the mean and standard deviation of the number of clusters in the Appendix of the revised version.

---

> > > > ### Comment · Reviewer_9SuC · 2025-08-05
> > > >
> > > > Actually, it is the first time I encounter such statistic concerning the estimated number of clusters. I think that is not informative enough. Why not reporting average and std?

---

> > > > > ### Author Response · Authors · 2025-08-05
> > > > >
> > > > > We apologize for the confusion. The AMI index measures the information-theoretic similarity between the clustering and the ground truth, while the ARI evaluates the pairwise correctness between the clustering and the ground truth. Both AMI and ARI are considered performance metrics.
> > > > >
> > > > > In contrast, the number of identified clusters is not a performance metric, as it does not provide insight into the quality of the clustering itself. However, in applied studies, it is common to report the number of clusters identified. For method comparisons, it's typically more informative to report AMI and ARI.
> > > > >
> > > > > Since we have already reported the mean and standard deviation of the AMI and ARI scores, we thought it would be useful to present the number of identified clusters differently. We intended to include box plots for the number of identified clusters in the revised version. However, since we cannot include these plots here, we have instead provided the mode of the number of identified clusters.
> > > > >
> > > > > That being said, the mean and standard deviations of the number of identified clusters are presented in the tables below. Since FADP1 and FADP2 are deterministic, running them multiple times on the same real dataset will always produce the exact same clustering results.
> > > > >
> > > > > | Datasets| FAEclust| FNN| FAE| VANO| funHDDC| funclust|FADP1|FADP2|
> > > > > |--|--|--|--|--|--|--|--|--|
> > > > > | BirdChicken (2)| 2.050 ± 0.224| 2.000 ± 0.000| 2.000 ± 0.000| 2.000 ± 0.000| 3.200 ± 0.834| 2.000 ± 0.000|2 |3 |
> > > > > | CBF (3)| 3.000 ± 0.000| 2.000 ± 0.000| 2.000 ± 0.000| 2.950 ± 0.224| 2.600 ± 0.503 | 2.100 ± 0.308|2 |3 |
> > > > > | Chinatown (2)| 2.000 ± 0.000 | 2.000 ± 0.000| 2.000 ± 0.000 | 2.100 ± 0.447| 4.400 ± 0.598| 2.050 ± 0.224 |2 |2 |
> > > > > | DSR (4)  | 5.950 ± 0.224 | 3.750 ± 0.444| 3.200 ± 0.410| 3.750 ± 0.444| 4.050 ± 0.224| 2.550 ± 0.510  |3 |2 |
> > > > > | ECG200 (2) | 2.050 ± 0.224| 2.050 ± 0.224| 2.050 ± 0.224| 2.000 ± 0.000 | 2.250 ± 0.444 | 2.300 ± 0.470 |2 |2 |
> > > > > | Fungi (18)| 15.90 ± 0.308| 6.850 ± 0.366| 7.850 ± 0.587 | 12.20 ± 0.523| 8.400 ± 0.503| 3.048 ± 1.071 |2 |5 |
> > > > > | Plane (7) | 7.000 ± 0.000 | 5.950 ± 0.224 | 6.000 ± 0.000 | 5.950 ± 0.224 | 5.250 ± 0.444| 2.600 ± 1.635  |4 |3 |
> > > > > | Rock (4) | 3.150 ± 0.366| 2.000 ± 0.000  | 2.000 ± 0.000  | 2.200 ± 0.410 | 3.150 ± 0.933| 4.000 ± 1.919 |2 |2 |
> > > > > | Symbols (6)  | 4.700 ± 0.470 | 4.000 ± 0.000 | 4.000 ± 0.000 | 4.050 ± 0.224 | 4.300 ± 0.470 | 2.450 ± 0.510 |3 |4 |
> > > > > | Blink (2) | 2.000 ± 0.000 | 2.000 ± 0.000 | 2.000 ± 0.000 | 2.000 ± 0.000| 2.200 ± 0.410 | 3.450 ± 0.887 |2 |2 |
> > > > > | BM (4) | 3.000 ± 0.000   | 3.000 ± 0.000 | 3.150 ± 0.366  | 3.000 ± 0.000 | 2.000 ± 0.000 | 4.700 ± 0.470 |2 |2 |
> > > > > | EOS (2) | 2.050 ± 0.224  | 2.000 ± 0.000 | 2.900 ± 0.308 | 2.000 ± 0.000 | 2.150 ± 0.366 | 2.250 ± 0.444|2 |2 |
> > > > > | Epilepsy (4) | 3.650 ± 0.489| 3.000 ± 0.000| 3.000 ± 0.000 | 3.000 ± 0.000| 7.050 ± 0.826  | 2.100 ± 0.308 |2 |3 |
> > > > > | ERing (6)  | 3.250 ± 0.444| 3.250 ± 0.444 | 3.800 ± 0.410 | 3.600 ± 0.503| 3.500 ± 0.513 | 2.150 ± 0.366|2 |4 |
> > > > > | FM (2)  | 4.150 ± 0.366  | 3.000 ± 0.324 | 3.000 ± 0.000 | 3.700 ± 0.470 | 2.400 ± 0.503 | 2.350 ± 0.587 |2 |6 |
> > > > > | JV (9) | 8.000 ± 0.000 | 2.300 ± 0.470 | 8.000 ± 0.000  | 8.000 ± 0.000 | 7.650 ± 0.489 | 2.250 ± 0.639 |2 |4 |
> > > > > | SWJ (3)| 2.300 ± 0.470 | 2.200 ± 0.410| 2.050 ± 0.224 | 2.350 ± 0.489 | 2.000 ± 0.000 | 2.000 ± 0.000|2 |2 |
> > > > > | | | | | | | | |
> > > > >
> > > > >
> > > > > | Datasets| FAEclust| FNN| FAE| VANO| funHDDC| funclust|FADP1|FADP2|
> > > > > |--|--|--|--|--|--|--|--|--|
> > > > > | Hypersphere (2) | 2.000 ± 0.000 | 6.150 ± 0.366| 3.950 ± 0.394 | 2.200 ± 0.410 | 4.100 ± 0.852| 2.900 ± 1.021 |2.000 ± 0.000 | 3.700 ± 1.302 |
> > > > > | Hyperpoblic (2) | 2.000 ± 0.000 | 4.050 ± 0.394 | 3.400 ± 0.503| 2.150 ± 0.366 | 5.350 ± 0.933| 2.100 ± 0.307|2.050 ± 0.224   | 3.250 ± 2.173 |
> > > > > | Swiss Roll (4) | 3.900 ± 0.308| 2.200 ± 0.410 | 2.200 ± 0.410| 2.700 ± 0.470 | 3.000 ± 0.561 | 2.450 ± 0.686|3.650 ± 1.089   | 2.100 ± 0.447 |
> > > > > | Lorenz (3) | 3.900 ± 0.308 | 3.850 ± 0.366 | 4.350 ± 0.489 | 2.100 ± 0.308 | 2.550 ± 0.510 | 2.900 ± 0.553 |2.050 ± 0.224 | 2.850 ± 1.089 |
> > > > > | Pendulum (4)| 4.000 ± 0.000| 2.000 ± 0.000 | 4.000 ± 0.000| 4.000 ± 0.000 | 4.000 ± 0.000 | 3.400 ± 0.503 |4.000 ± 0.000 | 4.000 ± 0.000 |
> > > > > | | | | | | | | |

---

### Official Review · Reviewer_CGH6 · 2025-07-04

**Clarity:** 4
**Significance:** 3
**Originality:** 3
**Rating:** 5
**Confidence:** 1

**Summary:**

This paper proposes a new unsupervised method to learn functional autoencoders for clustering multi-dimensional functional data. The new training objective applies to autoencoders, that involves a reconstruction loss, a clustering loss, and a regularization term on the functional weights and biases. The paper also discusses the forward and backward update of the clustering objective. The proposed method can handle functional data in both linear spaces and Riemannian manifolds. Experiments show that the proposed method outperforms existing state-of-the-art methods on learning time series data.

**Questions:**

1.	Is there an ablation study on the influence of the architecture on the learning? Namely, is your method sensitive to the choice of architecture?

**Ethical Concerns:**

["NO or VERY MINOR ethics concerns only"]

**Final Justification:**

The paper is well-written and easy to follow. I had some minor concerns to which the authors have addressed in their rebuttal. I am not an expert in the area as I have mentioned, so I cannot comment on the novelty and technique details. Suppose the novelty and technique is sound, then I would argue this is a fair contribution and deserves acceptance.

**Limitations:**

yes

**Quality:**

3

**Strengths And Weaknesses:**

**Strengths:**

1.	The paper is well-written and easy-to-follow. The notation was clear, which makes for great readability.
2.	I am not an expert of the area, so I am not in the position to comment on its novelty. But suppose the limitations of the previous methods for unsupervised learning of functional data were true, then the method is novel and sufficiently addresses the shortcomings of previous methods.
3.	Empirically, the proposed method seem to greatly outperforms existing methods.

 **Weaknesses:**

1.	Perhaps I missed the discussion in the paper, but it’s not clear how one can select the hyperparameter on the number of clusters. For example, L152 says “If an initial clustering of the FD is available”. In what cases will initial clustering of FD be available?

---

> ### Author Rebuttal · Authors · 2025-07-25
>
> **Response to Weaknesses:**
>
> Thank you for this observation. We agree that the discussion on determining the number of clusters should be made more prominent in the paper. Currently, this is only briefly mentioned in lines 314–315. We will revise the manuscript to clarify this point in multiple relevant sections. Below we provide a detailed explanation of our procedure:
>
> During the forward phase, we first update the embeddings and then apply our path-following homotopy algorithm to the new embeddings to obtain a hierarchy of agglomerative clustering solutions:
> 1. According to Eq. (5), for a given embedding $\pmb{X}$, the convex clustering objective has a unique solution for any value of $\lambda$.
> 2. As $\lambda$ increases, clusters merge one pair at a time, tracing out a full agglomerative hierarchy. Crucially, these merges happen only at specific values of $\lambda$ (termed breakpoints), such as $\lambda_K$, at which two clusters merge, yielding a partition of exactly K clusters (see lines 278–283). Between any two consecutive breakpoints (for example, when $\lambda_K\leq \lambda<\lambda_{K-1}$), the clustering remains exactly the same.
> 3. Our path-following homotopy algorithm efficiently identifies these breakpoints in O(n log(n)) time, thereby constructing the entire clustering hierarchy. This process can be further accelerated by initializing $\lambda$ with a warm-start value (lines 305-308).
>
> Once the hierarchy is constructed, we apply an internal validation index (e.g., silhouette score or Davies–Bouldin index) to select the optimal clustering result (and hence the number of clusters) from the hierarchy. Because this decision is driven entirely by the data, it does not depend on any particular $\lambda$ value; hence, the $\lambda$ in Eq. (5) is not a tunable hyperparameter.
>
> The network training process consists of several iterations of the forward-backward loop. In each iteration, we repeat the clustering and selection procedure described above. The final clustering result from the last iteration is reported as the output of the model.
>
> Regarding the mention of "an initial clustering of the FD," our intention was to convey that domain knowledge or simple and fast clustering algorithms (such as funHDDC) can be applied directly to the original functional data to obtain a rough initial estimate of cluster structure. While not required, such an initialization can help stabilize or accelerate training in practice. We will revise the text to better explain this and avoid potential confusion.
>
> **Response to Q1:**
>
> Thank you for the insightful question.  FAEclust extends a classical autoencoder architecture into the functional domain by introducing functional layers at both the input and output ends. Specifically:
>
> - Encoder: The first layer is a functional layer where functional weights (continuous functions) map the multi-dimensional functional input into a finite-dimensional latent space. After this, one or more fully connected MLP layers (with scalar weights) further process the latent representation.
> - Decoder: The decoder consists of three or more functional layers, each involving matrix multiplications with functional weights and biases followed by nonlinear activation functions (except for the final layer which is linear and has no bias). These layers map the latent code back into the space of functions, enabling reconstruction.
>
> Because FAEclust retains the same MLP core—depth, width, latent dimension, activation functions—as standard autoencoders, any architectural ablation study on these shared components would lead to similar conclusions to those already established in classical settings. (For example, Telgarsky 2016 [1] shows that additional depth and width exponentially expands representational power.) For this reason and to empirically validate the necessity of using multiple functional layers in the decoder, we focus our ablation study on the distinctive architectural component of FAEclust: the depth of the functional decoder. To this end, we compare two architectures:
>
> - a shallow variant with a single functional layer at the output end (architecture explained by the flow of information between lines 94 and 95 in the paper), and
>
> - the full version with three functional layers in the decoder (architecture explained by the flow of information between lines 109 and 110 in the paper).
>
> Both models share identical hyperparameter settings, where applicable. We ran each model 20 times per dataset and report the mean and standard deviation of AMI and ARI. The variation is due to the stochastic nature of Bayesian optimization and the mini-batch gradient descent algorithm. The results across the 17 real datasets are summarized below. It is evident that the three-layer decoder consistently improves clustering performance in terms of both AMI and ARI, often substantially. This highlights that deeper functional decoders not only align with theoretical guarantees but also yield tangible performance gains in practice.
>
> | Datasets | One-layer FAEclust |  &emsp;  Three-layer FAEclust |
> |--|--|--|
> |  | &emsp;&emsp; **AMI** &emsp;&emsp;&emsp;&emsp;&emsp; **ARI** | &emsp;&emsp; **AMI** &emsp;&emsp;&emsp; &emsp; &emsp; **ARI** |
> | BirdChicken | $0.295 \\pm 0.053$ &emsp; $0.239 \\pm 0.061$ | &emsp; $0.329 \\pm 0.037$ &emsp; $0.260 \\pm 0.052$ |
> | CBF | $0.483 \\pm 0.071$ &emsp; $0.465 \\pm 0.064$ | &emsp; $0.719 \\pm 0.043$ &emsp; $0.689 \\pm 0.054$ |
> | Chinatown | $0.194 \\pm 0.056$ &emsp; $0.170 \\pm 0.045$ | &emsp; $0.319 \\pm 0.024$ &emsp; $0.252 \\pm 0.020$ |
> | DSR | $0.769 \\pm 0.021$ &emsp; $0.752 \\pm 0.027$ | &emsp; $0.881 \\pm 0.029$ &emsp; $0.865 \\pm 0.041$ |
> | ECG200 | $0.117 \\pm 0.050$ &emsp; $0.136 \\pm 0.104$ | &emsp; $0.240 \\pm 0.035$ &emsp; $0.160 \\pm 0.065$ |
> | Fungi | $0.840 \\pm 0.042$ &emsp; $0.712 \\pm 0.063$ | &emsp; $0.926 \\pm 0.019$ &emsp; $0.847 \\pm 0.048$ |
> | Plane | $0.841 \\pm 0.043$ &emsp; $0.792 \\pm 0.067$ | &emsp; $0.904 \\pm 0.016$ &emsp; $0.826 \\pm 0.039$ |
> | Rock | $0.405 \\pm 0.019$ &emsp; $0.286 \\pm 0.025$ | &emsp; $0.455 \\pm 0.023$ &emsp; $0.309 \\pm 0.026$ |
> | Symbols | $0.567 \\pm 0.049$ &emsp; $0.410 \\pm 0.061$ | &emsp; $0.839 \\pm 0.031$ &emsp; $0.674 \\pm 0.039$ |
> | Blink | $0.516 \\pm 0.057$ &emsp; $0.470 \\pm 0.074$ | &emsp; $0.637 \\pm 0.038$ &emsp; $0.572 \\pm 0.043$ |
> | BM | $0.389 \\pm 0.102$ &emsp; $0.326 \\pm 0.116$ | &emsp; $0.542 \\pm 0.037$ &emsp; $0.493 \\pm 0.046$ |
> | EOS | $0.182 \\pm 0.019$ &emsp; $0.137 \\pm 0.027$ | &emsp; $0.261 \\pm 0.014$ &emsp; $0.206 \\pm 0.019$ |
> | Epilepsy | $0.255 \\pm 0.051$ &emsp; $0.192 \\pm 0.025$ | &emsp; $0.464 \\pm 0.039$ &emsp; $0.288 \\pm 0.034$ |
> | ERing | $0.586 \\pm 0.073$ &emsp; $0.454 \\pm 0.088$ | &emsp; $0.657 \\pm 0.047$ &emsp; $0.582 \\pm 0.053$ |
> | FM | $0.259 \\pm 0.045$ &emsp; $0.164 \\pm 0.056$ | &emsp; $0.261 \\pm 0.034$ &emsp; $0.180 \\pm 0.051$ |
> | JV | $0.723 \\pm 0.038$ &emsp; $0.651 \\pm 0.063$ | &emsp; $0.883 \\pm 0.016$ &emsp; $0.870 \\pm 0.020$ |
> | SWJ | $0.159 \\pm 0.064$ &emsp; $0.096 \\pm 0.080$ | &emsp; $0.322 \\pm 0.063$ &emsp; $0.257 \\pm 0.051$ |
> |  |  |  |
>
> [1] Matus Telgarsky. Benefits of depth in neural networks. 29th Annual Conference on Learning Theory, PMLR 49:1517-1539, 2016.

---

> > ### Comment · Reviewer_CGH6 · 2025-08-04
> >
> > Thank you for the clarification additional experiments. As I have stated in my review, I am not an expert in the area so I cannot comment too much on the novelty and what not. However, the rebuttal have answered my questions, and the added ablation and clarification can strengthen the paper.

---

### Decision · Program_Chairs · 2025-09-17

**Decision:**

Accept (poster)

**Comment:**

### Summary of scientific claims and findings

The paper introduces a functional autoencoder-based method for clustering multi-dimensional functional data in both Euclidean and Riemannian manifolds. The approach incorporates functional layers with basis function representations, a shape-informed clustering objective to address phase variation, and a convex clustering procedure. The authors prove a universal approximation theorem for the functional decoder and propose a homotopy-based algorithm for efficient clustering updates.

### Strengths

* **Clear contribution**: Extends deep autoencoders to functional data clustering with theoretical guarantees.
* **Novelty**: Introduces functional layers and a shape-informed clustering loss, which address limitations of existing functional clustering methods.
* **Breadth of evaluation**: Covers real, simulated, and manifold datasets, with systematic reporting of AMI and ARI across multiple repetitions.
* **Rebuttal improvements**: Added ablation studies, clarified the cluster selection procedure, re-implemented missing baselines, and reported robustness results.
* **Reproducibility**: Code package available with sensible defaults and documentation.

### Weaknesses and missing elements

* **Presentation issues**: Initial submission lacked pseudocode and clear explanations of some objectives, leading to confusion.
* **Hyperparameter complexity**: Method involves many hyperparameters, raising robustness and practicality concerns.
* **Novelty positioning**: While components are individually known (autoencoders, convex clustering, shape metrics), the distinct contribution relative to prior autoencoder-based works was under-emphasized initially.
* **Comparative evaluation**: Important baselines were missing in the original submission and only included after reviewer prompting.

### Key reasons for acceptance

1. The paper establishes a sound, well-motivated framework for clustering functional data, with both theoretical and empirical contributions.
2. The rebuttal successfully clarified earlier ambiguities and substantially strengthened the empirical evaluation by including missing baselines.
3. The work is novel in integrating functional layers, shape-informed clustering, and convex optimization into a unified deep clustering pipeline.

### Summary of discussion and rebuttal

**Before rebuttal**:

* `CGH6`: Positive on clarity and empirical performance but lacked domain expertise; questioned cluster selection procedure and architectural sensitivity.
* `9SuC`: Raised multiple concerns about novelty, clarity, hyperparameter tuning, and missing cluster-recovery details; leaned negative.
* `WLrd`: Questioned novelty positioning, missing baselines, and low manifold performance.
* `5zSQ`: Highlighting clarity, novelty, and design, but noted weak theory-to-practice connection and robustness concerns.

**Authors’ rebuttal**:

* Clarified cluster selection and hyperparameter tuning (no ground truth labels needed).
* Provided ablation on functional decoder depth.
* Re-implemented and added comparisons with missing baselines.
* Reported robustness via repeated runs and variance statistics.
* Addressed phase variation via elastic shape metric, showing clear benefits.

**Post-rebuttal positions**:

* `CGH6`: Maintained positive stance but deferred on novelty due to lack of expertise.
* `9SuC`: Raised to borderline accept but retained concerns about hyperparameters and reporting cluster numbers.
* `WLrd`: Increased score after additional experiments and baseline inclusion.
* `5zSQ`: Raised rating, satisfied with clarifications.

### Weighing these points

While initial reviews expressed substantial concerns, the rebuttal was strong and persuasive, addressing methodological clarity, missing baselines, and robustness. Remaining weaknesses relate mainly to presentation and complexity rather than to core soundness.

### Additional AC review

The integration into the current functional data literature appears a bit weak. Several papers on functional data were presented at last year's NeurIPS, but are not cited, nor are other functional machine learning approaches beyond clustering. I would encourage the authors to add these references.